# A dressed singlet-triplet qubit in germanium

K. Tsoukalas[1], U. von Lüpke[1], A. Orekhov [ORCID][1], B. Hetényi [ORCID][1], I. Seidler [ORCID][1], L. Sommer [ORCID][1], E. G. Kelly [ORCID][1], L. Massai [ORCID][1], M. Aldeghi [ORCID][1], M. Pita-Vidal [ORCID][1], N. W. Hendrickx [ORCID][1], S. W. Bedell [ORCID][2], S. Paredes [ORCID][1], F. J. Schupp [ORCID][1], M. Mergenthaler [ORCID][1], G. Salis [ORCID][1], A. Fuhrer [ORCID][1] & P. Harvey-Collard [ORCID][1] ✉

In semiconductor hole spin qubits, low magnetic field ($B$) operation extends the coherence time ($T_2^*$) but proportionally reduces the gate speed. In contrast, singlet-triplet (ST) qubits are primarily controlled by the exchange interaction ($J$) and can thus maintain high gate speeds even at low $B$. However, a large $J$ introduces a significant charge component to the qubit, rendering ST qubits more vulnerable to charge noise when driven. Here, we demonstrate a highly coherent ST hole spin qubit in germanium, operating at both low $B$ and low $J$. By modulating $J$, we achieve resonant driving of the ST qubit, obtaining an average gate fidelity of 99.68% and a coherence time of $T_2^* = 1.9\,\mu$s. Moreover, by applying the resonant drive continuously, we realize a dressed ST qubit with a tenfold increase in coherence time ($T_{2\rho}^* = 20.3\,\mu$s). Frequency modulation of the driving signal enables universal control, with an average gate fidelity of 99.63%. Our results demonstrate the potential for extending coherence times while preserving high-fidelity control of germanium-based ST qubits, paving the way for more efficient operations in semiconductor-based quantum processors.

Among the various semiconductor spin qubit platforms[1], holes in germanium have gained significant attention due to their unique advantages. These include the absence of valley states[2], the inherent spin-orbit coupling that enables all-electrical control[3], and the formation of high-quality two-dimensional quantum dot (QD) arrays[4,5]. While spin-orbit interaction facilitates fast electrical drive (through g-tensor modulation), it also increases the coupling to charge fluctuations, limiting the qubit coherence time[6–8]. A potential solution is to operate at low in-plane magnetic fields ($B$) to reduce the effect of spin-orbit coupling and thus extend the dephasing time[9]. However, this comes at the cost of proportionally slowing down qubit drive speed, resulting in no clear advantage in single-qubit gate fidelity[9].

In contrast, gates between two spins are typically generated by the exchange interaction, which is independent of $B$, hence enabling fast operations even at low magnetic field[5]. By using a singlet-triplet qubit, the exchange interaction can also serve as a driving mechanism for single-qubit control[10–12].

The singlet-triplet (ST) qubit discussed here is encoded in the antipolarized states ($S_z = 0$) of two spins in a double QD with a Zeeman energy difference of $\Delta E_Z$[10]. Single-qubit operations can be achieved by diabatically tuning the exchange interaction ($J$)[13]. However, this simple implementation has the drawback of non-orthogonal control axes arising from the combination of $J$ and the non-tunable $\Delta E_Z$. Meanwhile, achieving $J \gg \Delta E_Z/h$ is in general difficult and makes the qubit more susceptible to charge noise. An efficient alternative control mechanism for ST qubits is resonant exchange driving[10]. This approach has the potential to preserve the dephasing times of ST qubits in two ways. First, the exchange interaction is kept at a moderate value and is modulated at the ST qubit frequency[14–16]. This enables resonant driving while maintaining detuning at the symmetric operation point, decreasing sensitivity to charge noise[17]. Second, resonant driving mechanisms, such as electron spin resonance, have been shown to decouple spin qubits from certain noise frequency components, achieving orders-of-magnitude longer dephasing times[18]. This has led to the development of continuously-driven (or 'dressed') spin qubits, where record-high coherence times and high-fidelity gates have been achieved[19–21].

The application of such dressing techniques to ST qubits operated at low magnetic fields remains unexplored, and offers a path towards

[1]IBM Research Europe – Zurich, Rüschlikon, Switzerland. [2]IBM Quantum, T.J. Watson Research Center, Yorktown Heights, NY, USA.
✉e-mail: phc@zurich.ibm.com

longer dephasing times and high-fidelity control. This is particularly compelling because dressed qubits benefit from large Rabi frequencies, which can be efficiently realized through resonant exchange driving. Notably, a recent demonstration of multiple ST qubits in germanium highlights the potential of ST qubit systems[12].

In this work, we study the performance of a dressed ST qubit with holes in Ge and compare it to the bare resonantly-driven ST qubit. The magnetic field is fixed at 20 mT to balance the effects of spin-orbit interaction on spin relaxation, dephasing, and driving. We first implement universal single-qubit control of a resonantly-driven ST qubit at the symmetric operation point, and characterize the gate fidelity with randomized benchmarking. We then proceed to dress the ST qubit with a continuous drive, implement two-axis control using frequency modulation (FM) of the resonant exchange drive, and benchmark again the single qubit operations. We obtain very similar gate fidelities of 99.68(2)% and 99.63(7)% with $X_\pi$ gate durations of 327 ns and 500 ns for the bare and dressed qubits, respectively, while the dressed qubit coherence time is improved by an order of magnitude.

## Results
### Device and measurement protocol
The device, shown in Fig. 1a, comprises a six-QD array defined electrostatically in the quantum well of a Ge/SiGe heterostructure[2,22,23]. A double-layer gate structure is employed, first layer barrier gates (B, teal), and second layer plunger gates (P, blue), with each Ti/Pd layer isolated by a conformal SiO$_2$ layer as depicted in the schematic in Fig. 1a(inset). The hole reservoirs are made of PtSiGe formed by annealing a thin platinum layer[24]. The charge occupation of each QD is detected by the change in the current ($I_\text{sensor}$) passing through a nearby sensing QD[25]. All measurements have been performed in a dilution refrigerator at a base temperature of 20 mK and with an external magnetic field of $B = 20$ mT parallel to the surface of the device.

By monitoring $I_\text{sensor}$ while changing the voltages of the two virtual gates[26] vP1 and vP2 (Supplementary Information (SI) Section G), we measure the charge stability diagram shown in Fig. 1b. Charges load into QD1 (under P1) directly from the reservoir O1, while for QD2 (under P2) the loading is slow and happens through QD1, resulting in latching effects. We operate the device in the (1,1) charge configuration where we initialize the spins in their ground state ($|\downarrow\downarrow\rangle$). This is done by

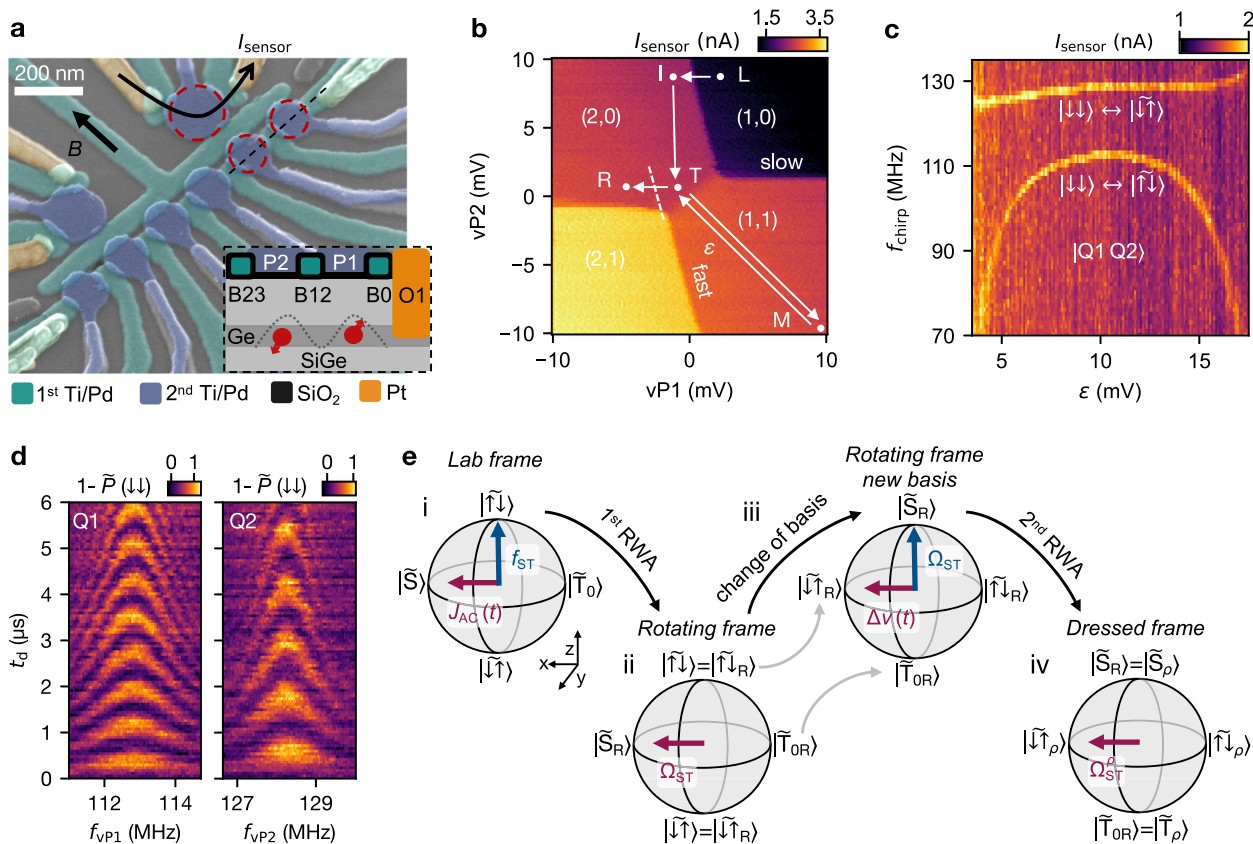

**Fig. 1 | Germanium hole ST qubit device. a** A false-colored tilted scanning electron microscope image of a nominally identical six QD device to the one used in the experiments. Inset: Schematic of the device cross section along the dotted black line in (**a**). **b** Double QD charge stability diagram as a function of vP1 and vP2 with the different charge configurations in QD1 and QD2 indicated by ($N_1$, $N_2$). The points and arrows describe the pulse sequence used in the experiment. L: Reset with one hole loaded in QD1. I: Loading of second hole and initialization of the two-spin system in the S(2,0) state. T: Turning point before and after crossing the interdot transition. M: Manipulation point where all control pulses are applied. With $\varepsilon$ we denote the detuning from the interdot in units of its vP1 coordinate. R: The latched readout point, resulting in the $|\downarrow\downarrow\rangle$ state being distinguished from all other spin states $|\widetilde{\downarrow\uparrow}\rangle$, $|\widetilde{\uparrow\downarrow}\rangle$, $|\uparrow\uparrow\rangle$. **c** Spectroscopy of transitions from the $|\downarrow\downarrow\rangle$ state inside the (1,1) region as a function of $\varepsilon$, measured with a chirped pulse applied to vP2 around

the frequency ($f_\text{chirp}$). Two transition lines are visible that are interpreted as the $|\downarrow\downarrow\rangle \leftrightarrow |\widetilde{\downarrow\uparrow}\rangle$ and $|\downarrow\downarrow\rangle \leftrightarrow |\widetilde{\uparrow\downarrow}\rangle$ transitions. **d** Rabi chevron patterns of Q1 and Q2 with the drive signals applied to vP1 and vP2, respectively. $\bar{P}$ is calibrated using the average current values in the (2,1) and (1,1) regions. **e** Step-by-step illustration of the dressing procedure for the ST qubit, depicted through Bloch spheres at each stage. (i) Resonantly driven ST qubit in the lab frame, with a frequency $f_\text{ST}$ and a drive term $J_\text{AC}(t)$. (ii) Applying a rotating wave approximation (RWA) and transitioning to the rotating frame removes the time dependence of $J_\text{AC}$ along with $f_\text{ST}$. (iii) Transformed basis states where $\Omega_\text{ST}$ points along the poles. The driving term $\Delta\nu$ emerges from either a second tone or a detuning in $\Omega_\text{ST}$. (iv) The application of a second RWA results in the dressed ST frame where the time dependence of the new drive along with $\Omega_\text{ST}$ have been removed.

first preparing a S(2,0) state, followed by a slow ramp of the detuning $\varepsilon$ (see pulse sequence in Fig. 1c) into the (1,1) region, ensuring adiabatic passage of the S(2, 0) $\leftrightarrow$ T$_-$(1, 1) anticrossing (SI Section A).

The spin state is read out using a latched Pauli spin blockade (PSB) technique. Fully adiabatic spin-to-charge conversion first maps the (1,1) ground state $|\downarrow\downarrow\rangle$ to S(2,0), while all other (1,1) excited states ($|\widetilde{\downarrow\uparrow}\rangle$, $|\widetilde{\uparrow\downarrow}\rangle$, $|\uparrow\uparrow\rangle$) remain blockaded. Note that the tilde indicates the hybridization of antipolarized spin states due to the finite residual exchange coupling ($J \approx 6$ MHz). Then, by pulsing to point R, the blockaded states are converted to the (2,1) charge state while the singlet state remains in (2,0), resulting in both signal and lifetime enhancements[27,28].

We find the two spin flip transition frequencies corresponding to $|\downarrow\downarrow\rangle \leftrightarrow |\widetilde{\downarrow\uparrow}\rangle$ and $|\downarrow\downarrow\rangle \leftrightarrow |\widetilde{\uparrow\downarrow}\rangle$ by applying a broadband (chirped) voltage drive to vP2 around frequency $f_{\text{chirp}}$. In Fig. 1c, we show the measured energy spectrum of the two-spin system as a function of the detuning $\varepsilon$. The operation point is set to the symmetric point in the middle of (1,1) ($\varepsilon = 10$ mV, point M in Fig. 1c), where the system is the least sensitive to variations in detuning[17]. At this point, the transition frequencies of the spins are $f_{Q1} = 112.9$ MHz (g-factor of 0.41) and $f_{Q2} = 128.23$ MHz (g-factor of 0.44).

By switching to single-tone driving signals at frequencies $f_{vP1}$ and $f_{vP2}$ on the gates vP1 and vP2, respectively, we observe a Rabi chevron pattern for each of the two qubits (see Fig. 1d). With the given heterostructure and $B$ orientation, the driving mechanism is expected to be g-tensor magnetic resonance (g-TMR)[8,9,29]. We then calibrate the timing of single-qubit $X_\pi^{Q1(2)}$ gates for Q1(2) following the procedure described in the SI Section B. The $|\widetilde{\uparrow\downarrow}\rangle$ initialization procedure used for the ST qubit consists of first adiabatically converting a S(2,0) into (1,1)$|\downarrow\downarrow\rangle$, followed by a flip of Q1. Similarly, the $|\widetilde{\uparrow\downarrow}\rangle$ readout procedure reverses these steps in the opposite order, effectively distinguishing $|\widetilde{\uparrow\downarrow}\rangle$ from the other three states (see SI Section B.). The time-resolved sensor current is integrated over 100 $\mu$s at the R point, and then averaged over 1000 experimental shots. The resulting average $I_{\text{sensor}}$ is converted to an approximate probability scale $\tilde{P}$ and the method used to calibrate the scale is explained in the captions of the figures of the respective measurements.

## Resonantly-driven ST qubit

We proceed to implement the resonantly-driven ST qubit. We describe it with a Hamiltonian defined in the antipolarized subspace $\{|\widetilde{\downarrow\uparrow}\rangle, |\widetilde{\uparrow\downarrow}\rangle\}$:

$$H_{\text{res}}^{\text{ST}}/h = \frac{1}{2}(f_{\text{ST}}\sigma_z + J_{\text{AC}}(t)\sigma_x), \qquad (1)$$

where $f_{\text{ST}} = \sqrt{(\Delta E_Z/h)^2 + J_{\text{DC}}^2}$, $\Delta E_Z$ is the Zeeman energy difference between the two spins, and $J_{\text{DC}}$ is the dc exchange interaction ($J_{\text{DC}} = 6$ MHz at $\varepsilon = 10$ mV). $J_{\text{AC}} = A_J \cos(2\pi f_d t)$ is the component of the modulated exchange along $\sigma_x$ (the one along $\sigma_z$ is neglected), as defined in Fig. 1e(i), where $A_J$ is the amplitude and $f_d$ the drive frequency.

Applying a drive tone to vB12 with frequency $f_d$ and amplitude $A$ results in $J_{\text{AC}} \neq 0$. When $f_d$ matches $f_{\text{ST}} = 15.65$ MHz, resonant driving takes place, allowing for rotations around $J_{\text{AC}}$ in the Bloch sphere with a Rabi frequency $\Omega_{\text{ST}}$. To describe this, we move to the frame rotating with $f_{\text{ST}}$ where the basis states are $|\widetilde{\uparrow\downarrow}_R\rangle$ and $|\widetilde{\downarrow\uparrow}_R\rangle$ as shown in Fig. 1e(ii). In Fig. 2a, we show the characteristic Rabi chevron pattern emerging from sweeping $f_d$ in a range around $f_{\text{ST}}$. After locating the frequency of the transition, we calibrate the $X_\pi^{\text{ST}}$ and $X_{\pi/2}^{\text{ST}}$ pulses for the resonant singlet-triplet qubit following the procedure described in the SI Section B.

At the M point, the exchange depends approximately exponentially on the barrier voltage, leading to a driving term $J_{\text{AC}} \propto \exp(\text{vB12}) \sim \exp(A \sin(2\pi f_d t))$[30]. This behavior is observed in Fig. 2b, where the dependence of the Rabi frequency $\Omega_{\text{ST}}$ on the voltage amplitude $A$ is plotted. From the Fourier transform (Fig. 2b inset),

we observe the non-linear relation between the Rabi frequency and $A$, a behavior also observed in the supporting simulations in the SI Section C.

In Fig. 2c, we demonstrate two-axis control with the resonantly-driven ST qubit by adjusting the drive phase $\phi$. Starting from $|\widetilde{\uparrow\downarrow}\rangle$ and applying an $X_{\pi/2}^{\text{ST}}$ pulse, we place the system in a state pointing along the Y axis of the Bloch sphere of Fig. 1e(ii). Subsequently, we apply a Rabi drive pulse of duration $t_d$ and a varying phase $\phi$, which determines the rotation axis in the $xy$ plane, before we read out the population of the $|\widetilde{\uparrow\downarrow}\rangle$ state. For $\phi = \pm \pi/2$, we observe no Rabi oscillations, with the state probability remaining at $\simeq 50\%$. This corresponds to rotations around the Y axis (Y$^{\text{ST}}$ gates), which, combined with rotations around the X axis (X$^{\text{ST}}$ gates), allow full access to the ST qubit Bloch sphere. In addition, Z gates can be implemented by computer-added phase to the subsequent pulses[7].

To estimate the single-qubit gate fidelity of the resonantly-driven ST qubit, we perform randomized benchmarking (RB) using the Clifford gate set $\{X_\pi^{\text{ST}}, X_{\pi/2}^{\text{ST}}, Z_\pi^{\text{ST}}, Z_{\pi/2}^{\text{ST}}, I^{\text{ST}}\}$, applying gate sequences of up to 485 Clifford gates and 100 randomizations. The $X_\pi^{\text{ST}}$ and $X_{\pi/2}^{\text{ST}}$ gate times are 327 ns and 165 ns, respectively, while $Z_\pi^{\text{ST}}, Z_{\pi/2}^{\text{ST}}, I^{\text{ST}}$ are considered instantaneous and error-free. We fit the resulting data to the function $\alpha + \beta p^{n_C}$, where $\alpha$ is an offset constant, $\beta$ the visibility, $p$ the depolarization parameter and $n_C$ the number of Clifford gates. The average error per Clifford is calculated as $r_C = (1 - p)/2$[31]. On average, the number of gates per Clifford gate is 2.125, and the fraction of physical gates is 0.392 (not counting I and Z gates as physical). From the RB data shown in Fig. 2d, we extract an average physical gate fidelity of $\mathcal{F} = (1 - r_C/(2.125 \times 0.392)) = 99.68(2)\%$, which is among the highest reported for a ST qubit[12,16,32].

We complete the characterization of the resonantly driven ST qubit with a measurement of the key decay times. By applying the pulse sequences described in Fig. 2e, we find a dephasing time of $T_2^* = 1.9 \mu$s and a Hahn echo time of $T_2^H = 4.2 \mu$s. Both coherence times are among the highest reported for hole ST systems[11,12,33]. Although operation at low $B$ increases the dephasing time, we note that there is a trade-off between dephasing time and gate speed (for resonant gates), which can be fundamentally improved by operating in specific $B$ field directions[9]. A Rabi oscillation is plotted in Fig. 2f from which a Rabi decay time of $T_2^R = 20.3 \mu$s is extracted, constituting an order of magnitude longer than $T_2^*$. Finally, regarding the decay time $T_1$, no decay was observed during a wait time of 5 ms, hence we lower bound $T_1 > 5$ ms. The simple electrical control of this qubit, the high Rabi frequency achievable, and the long $T_1$ time make it an excellent candidate for implementing continuous driving protocols.

## Dressed ST qubit state space

The continuous application of a resonant exchange drive leads to the realization of a dressed ST qubit. To understand this qubit's state space, it is useful to first change the basis states of the rotating frame from $\{|\widetilde{\uparrow\downarrow}_R\rangle, |\widetilde{\downarrow\uparrow}_R\rangle\}$ in Fig. 1e(ii) to $\{|\widetilde{S}_R\rangle, |\widetilde{T}_{0R}\rangle\}$ in Fig. 1e(iii).

The Hamiltonian describing the dynamics is:

$$H_R^{\text{ST}}/h = \frac{1}{2}(\Omega_{\text{ST}}\tau_z + \Delta\nu(t)\tau_x), \qquad (2)$$

where $\tau_{x,y,z}$ are the Pauli matrices defined in the $\{|\widetilde{S}_R\rangle, |\widetilde{T}_{0R}\rangle\}$ basis (see the derivation in SI Section D). The presence of the off-diagonal term $\Delta\nu(t)$ enables various driving techniques for dressed qubits[18]. In this work, we investigate two techniques, both of which generate an oscillating $\Delta\nu(t)$. When the frequency of $\Delta\nu(t)$ equals $\Omega_{\text{ST}}$, the resonant driving condition is satisfied, leading to state rotations around $\tau_x$ with a Rabi frequency of $\Omega_{\text{ST}}^\rho$. We illustrate this with a Bloch sphere defined in the dressed frame, where the basis states are $|\widetilde{S}_\rho\rangle$ and $|\widetilde{T}_{0\rho}\rangle$, as shown in Fig. 1e(iv).

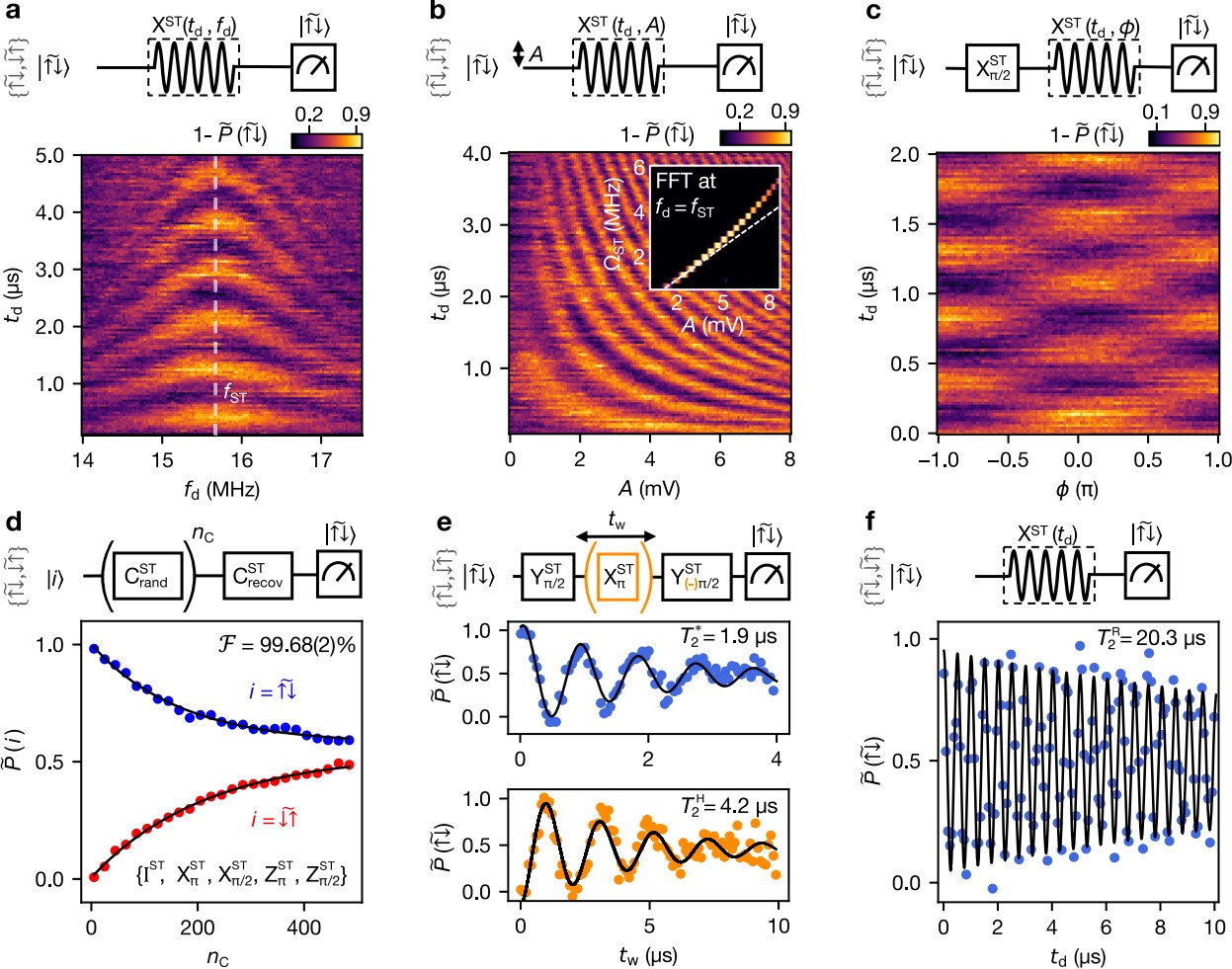

**Fig. 2 | Resonantly-driven singlet-triplet qubit. a** Rabi chevron pattern with a drive signal on vB12 of duration $t_d$ and frequency $f_d$. Initialization and readout of the $|\tilde{\uparrow}\downarrow\rangle$ state is described in the main text. $\tilde{P}$ is normalized with the application of either an $I^{Q1}$ or an $X_\pi^{Q1}$ before the measurement sequence. Above each plot, the corresponding pulse sequence or circuit diagram is illustrated. **b** Dependence of the Rabi frequency, $\Omega_{ST}$, on the drive amplitude $A$. The divergence from linearity (white dashed line) seen in the fast Fourier transform (inset) stems from the exponential dependence of the exchange to vB12. **c** Demonstration of rotation axis control via the addition of a phase $\phi$. An $X_{\pi/2}^{ST}$ pulse initializes the system in a state pointing along the Y axis of the Bloch sphere. The Rabi oscillations disappear for $\phi = \pm\pi/2$, indicating that the drive axis and state align. **d** Randomized benchmarking of the single-qubit gates in the singlet-triplet subspace. We extract an average gate fidelity of $\mathcal{F} = 99.68(2)\%$. **e** Ramsey (above) and Hahn echo (below) experiments fitted to a decaying exponential, with $T_2^* = 1.9\,\mu s$ and $T_2^H = 4.2\,\mu s$ respectively. **f** Exponential decay fit of the Rabi oscillation with $T_2^R = 20.3\,\mu s$.

## Dressed ST qubit spectroscopy

The first technique we investigate is the two-tone drive, demonstrated in Fig. 3a, which we use to perform spectroscopy of the dressed ST qubit[18]. A first pump signal, at frequency $f_{pump} = f_{ST}$ and with varying amplitude $A_{pump}$, is responsible for dressing the ST qubit. At the same time, a second weaker probe signal ($A_{probe} = 0.3$ mV) with frequency $f_{probe}$ swept around $f_{ST}$ induces transitions of the dressed qubit. In this configuration, the drive term $\Delta\nu(t) \propto A_{probe}\cos(2\pi\Delta f\,t)$, where $\Delta f = f_{probe} - f_{ST}$ (see Section D of the SI).

The measurement results, shown in Fig. 3a, reveal background Rabi oscillations of the ST qubit, along with three prominent features where the oscillations are perturbed. These include a central feature at $\Delta f = 0$, corresponding to an increase in the effective Rabi drive amplitude when $f_{probe} = f_{ST}$. To explain the two additional features symmetrically spaced around $f_{ST}$, we refer to the Bloch sphere shown in Fig. 1e(iii). When $\Delta f = \pm\Omega_{ST}$, resonant driving in the dressed qubit subspace $\{|\tilde{S}_\rho\rangle, |\tilde{T}_{0\rho}\rangle\}$ takes place, as depicted in Fig. 1e(iv), leading to spin rotations. This effect is also known as the "Mollow triplet" in the context of the Jaynes-Cummings Hamiltonian[18,34]. Its frequency

dependence on $A_{pump}$ reflects the non-linear relation between the voltage amplitude and the exchange, as discussed in Fig. 2b. The full theoretical derivation along with simulations of the two-tone spectroscopy experiment can be found in Section D of the SI. For the experiments shown in the following section, the ST qubit is continuously driven with $\Omega_{ST} = 4.2$ MHz.

## Dressed ST qubit control

We now demonstrate universal control of the dressed ST qubit by dynamically detuning the frequency of the driving signal $f_d$ away from $f_{ST}$. This results in an off-diagonal term given by $\Delta\nu(t) = f_d(t) - f_{ST}$ and is depicted in the Bloch sphere in Fig. 1e(iii).

The frequency modulation (FM) technique consists in periodically modulating $f_d(t)$ around $f_{ST}$. This leads to a drive term $\Delta\nu(t) = \Delta\nu_{FM}\cos(2\pi f_{FM}\,t + \phi_{FM})$ with a characteristic FM frequency $f_{FM}$ and amplitude $\Delta\nu_{FM}$ (SI Section D). When $f_{FM} = \Omega_{ST}$, the application of a second RWA results in the dressed qubit Bloch sphere shown in Fig. 1e(iv) where, similarly to the two-tone case, the state rotates around $\tau_x$ with a frequency $\Omega_{ST}^\rho = \Omega_{ST}^{FM}$.

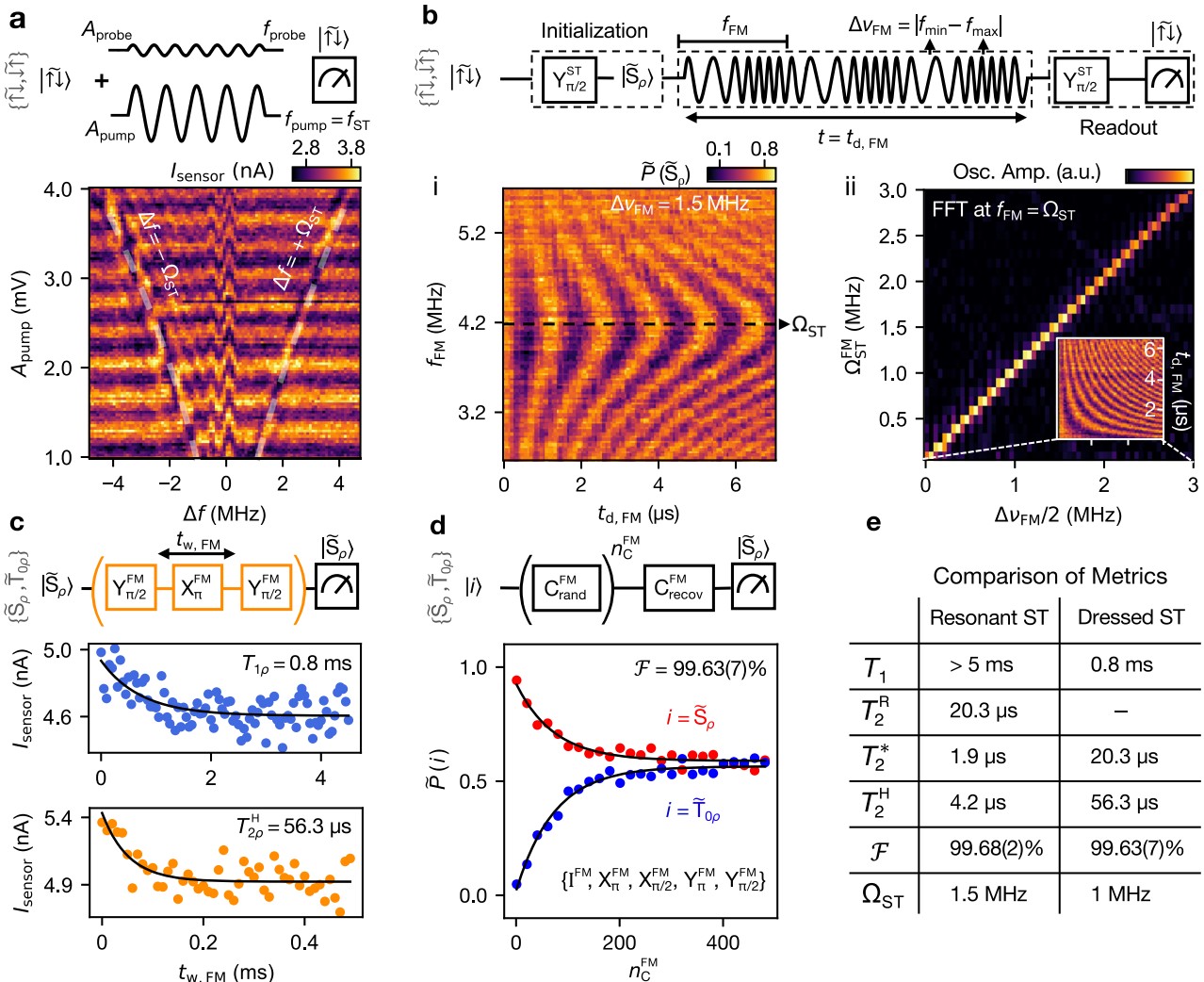

**Fig. 3 | Dressed singlet-triplet qubit via resonant exchange. a** Two-tone spectroscopy of the dressed ST qubit with a pump and a probe signal applied for a duration of $3\,\mu s$. Varying the pump signal amplitude ($A_{pump}$) results in Rabi oscillations in the $\{|\widetilde{\uparrow\downarrow}\rangle, |\widetilde{\downarrow\uparrow}\rangle\}$ subspace. When the frequency of the probe signal $f_{probe}$ is resonant with $f_{ST}$ or $f_{ST} \pm \Omega_{ST}$, the measured Rabi oscillation pattern is disrupted. The two signals have a phase difference of $\pi/4$. This choice was made to achieve maximum visibility of all three branches. Above each plot, the corresponding pulse sequence or circuit diagram is illustrated. **b** A frequency modulated driving signal gives rise to an oscillating (with frequency $f_{FM}$) driving term that enables rotations in the dressed ST subspace. The state $|\widetilde{S}_\rho\rangle$ is initialized, then rotated by FM modulation of the drive signal before being projected to $|\widetilde{T}_{0\rho}\rangle$ for readout. (i) FM driven Rabi chevron pattern. (ii) Fourier transform of the amplitude ($\Delta\nu_{FM}$) dependence of the dressed Rabi frequency $\Omega_{ST}^{FM}$ (inset), plotted in arbitrary units (a.u.). **c** (Top) Measurement of the dressed qubit decay time $T_{1\rho} = 0.8$ ms. (Bottom) Hahn echo experiment to estimate the dressed qubit echo coherence time $T_{2\rho}^H = 56\,\mu s$, where $t_{w,FM}$ is the free evolution time not including the duration of $X_\pi^{FM}$. **d** A fidelity of $\mathcal{F} = 99.63(7)\%$ is extracted for the FM dressed qubit gates via randomized benchmarking performed for both $|\widetilde{S}_\rho\rangle$ and $|\widetilde{T}_{0\rho}\rangle$ initial states. **e** Comparison between the performance metrics of the resonantly-driven and dressed ST qubits.

To initialize and read out the dressed qubit, we leverage the fact that $|\widetilde{S}_\rho\rangle = |\widetilde{S}_R\rangle$ (and $|\widetilde{T}_{0\rho}\rangle = |\widetilde{T}_{0R}\rangle$). Hence, by applying an additional $Y_{\pi/2}^{ST}$ pulse, we rotate the $|\widetilde{\uparrow\downarrow}\rangle$ state to $|\widetilde{S}_\rho\rangle$ to initialize (and the $|\widetilde{T}_{0\rho}\rangle$ state to the $|\widetilde{\uparrow\downarrow}\rangle$ state to read out), before continuously driving the frequency modulated signal, as illustrated in the pulse schematic in Fig. 3b.

In Fig. 3b(i), we show the measured Rabi chevron pattern that emerges when $f_{FM}$ is scanned around $\Omega_{ST}$. Resonant driving occurs at $f_{FM} = \Omega_{ST} = 4.2$ MHz, with a corresponding Rabi frequency of $\Omega_{ST}^{FM} = 0.75$ MHz for $\Delta\nu_{FM} = 1.5$ MHz. The timings of the $X_\pi^{FM}$ and $X_{\pi/2}^{FM}$ gates are calibrated following the same procedure as before. Similarly to the resonantly-driven ST case in Fig. 2c, a phase $\phi_{FM}$ can be introduced in the FM signal, allowing for rotations around the Y-axis of the dressed Bloch sphere (see SI Fig. S5).

A key characteristic of the FM driving is that the driving amplitude $\Delta\nu_{FM}$ is a frequency bandwidth, with $\Omega_{ST}^{FM} = \Delta\nu_{FM}/2$. We measure $\Omega_{ST}^{FM}$ as a function of $\Delta\nu_{FM}$, as shown in Fig. 3b(ii). The fast Fourier transform

reveals the linear 1:1 relationship between $\Delta\nu_{FM}/2$ and $\Omega_{ST}^{FM}$[35]. By increasing $\Delta\nu_{FM}/2$ to values close to or larger than $\Omega_{ST}^{FM}$ (see SI Section F), the system enters a regime where the RWA ceases to be valid[35]. Additionally, in this scheme, increasing the drive amplitude does not necessitate a higher voltage, which could cause heating or crosstalk[36,37], but rather a broader modulation bandwidth $\Delta\nu_{FM}$, limited only by the capabilities of the control electronics[18].

After calibrating the single qubit gates, we set $\Delta\nu_{FM} = 2$ MHz, and execute more complex pulse sequences to extract relevant metrics for the system. First, the dressed decay time $T_{1\rho}$ is measured by initializing in the $|\widetilde{S}_\rho\rangle$ state, driving the system for a time $t_{w,FM}$, and then reading out the $|\widetilde{S}_\rho\rangle$ population. By fitting the data to an exponential decay, we find a decay time $T_{1\rho} = 0.8$ ms. This is lower than in the resonantly-driven ST qubit case, as expected for dressed qubits[38], yet sufficiently long to not limit any coherent processes. The Ramsey sequence for the dressed qubit coincides with the Rabi drive sequence of the resonantly-driven ST qubit but is interpreted in the dressed frame, resulting in

$T_{2\rho}^* = T_2^R = 20.3\,\mu s$. Subsequently, by inserting an $X_\pi^{FM}$ echo pulse between the initialization and readout of the $|\uparrow\downarrow\rangle$ state, as depicted in the schematic in Fig. 3c, the state is refocused and a Hahn echo time of $T_{2\rho}^H = 56\,\mu s$ is extracted. The introduction of more refocusing pulses further increases the coherence time, as shown in the SI Fig. S5c.

To characterize the single-qubit gate performance of the dressed qubit, we again perform randomized benchmarking. The benchmarking sequence is applied with both $|\widetilde{S}_\rho\rangle$ and $|\widetilde{T}_{0\rho}\rangle$ states as inputs, to eliminate readout drift errors. We apply a maximum of 481 Clifford gates ($n_C^{FM}$) from the set $\{X_{\pm\pi/2}^{FM}, X_\pi^{FM}, Y_{\pm\pi/2}^{FM}, Y_\pi^{FM}, I^{FM}\}$ (Section IVB). Figure 3d shows the measured sequence fidelity decay as a function of the number of Clifford gates applied. We extract an average physical gate fidelity of $\mathcal{F} = 99.63(7)\%$, with 1.875 gates per Clifford, and noting that all gates except $I^{FM}$ (do nothing for no time) in the set are physical. This fidelity is, within error bars, comparable to that of the resonant ST qubit gates.

## Discussion

In summary, we realized and characterized a resonantly driven singlet-triplet and a dressed singlet-triplet hole qubit in germanium. A comparison of their key metrics is presented in the table in Fig. 3e. By resonantly driving the exchange interaction at the detuning symmetry point, we achieve high-fidelity ST qubit rotations (99.68(2)%) with good coherence times ($T_2^* = 1.9\,\mu s$). Continuous driving helps decouple the qubit from environmental noise, resulting in an order-of-magnitude increase in coherence time. Furthermore, qubit rotations in the dressed system, implemented via a frequency-modulated driving signal, show a fidelity of 99.63(7)%, which is within the error margin compared with the bare ST qubit.

Dressing the qubit has improved fidelity in some cases (e.g., Vallabhapurapu et al.[39]) but not in others (e.g., Hansen et al.[20]). The lack of fidelity improvement in our work could be related to both unitary and decoherence errors. Unitary errors could be introduced by the high drive speed used to reduce the gate duration, challenging the RWA. A reduction of decoherence errors can be achieved by an additional improvement of the noise filtering through the application of more elaborate dressing schemes[20] and reduced nuclear spin noise[9]. Even without improvement in the gate fidelity, the advantage of the dressed scheme lies in the extended coherence time, which is especially beneficial during idle periods such as readout or two-qubit gate operations elsewhere in the device. Compared with CPMG, the dressing approach offers better synchronicity and is simple to implement. In the future, the drive signal on the barrier could be transformed to compensate for the exponential dependence of the exchange interaction on the voltage, suppressing the higher order terms[40].

These findings demonstrate, on the one hand, the highly coherent nature of dressed ST hole qubits at low magnetic fields, and on the other hand, the effectiveness of resonant exchange as a driving mechanism. A logical next step involves the implementation of two-qubit gates between dressed ST qubits. Recent developments in two-qubit interaction schemes, for both ST qubits[12] and dressed spin-1/2 qubit systems[21], offer valuable insights for achieving entangling operations in dressed ST qubits. By harnessing the resonant exchange drive, the dressed ST qubit has the potential to play an important role in semiconductor quantum technologies.

## Methods
### Experimental Setup
All measurements are performed in a Bluefors XLD dilution refrigerator with a base temperature of 20 mK. The sample is mounted on a QDevil QBoard circuit board. Static gate voltages are applied via a QDevil QDAC through DC lines, which are filtered at the millikelvin stage using a QDevil QFilter. All plunger and barrier gates are connected to coaxial lines through on-PCB bias tees. RF lines are attenuated by 20 dB inside the refrigerator. Fast voltage excitation pulses are delivered to the quantum dot gates using Tektronix AWG5204

**Table 1 | Clifford gate implementations for resonantly driven and dressed ST qubits**

| Clifford Gates | |
|---|---|
| **Resonant** | **Dressed** |
| I | I |
| − Z, X | Y/2, X/2 |
| − Z, X/2 | − X/2, − Y/2 |
| − Z, X/2, − Z | X |
| − Z, X/2, − Z/2 | − Y/2, − X/2 |
| − Z, X/2, Z/2 | X/2, − Y/2 |
| − Z/2 | Y |
| − Z/2, X/2 | − Y/2, X/2 |
| − Z/2, X/2, − Z | X/2, − Y/2 |
| − Z/2, X/2, − Z/2 | X, Y |
| − Z/2, X/2, Z/2 | Y/2, − X/2 |
| X | − X/2, Y/2 |
| X, Z/2 | Y/2, X |
| X/2 | − X/2 |
| X/2, − Z | X/2, − Y/2, − X/2 |
| X/2, − Z/2 | − Y/2 |
| X/2, Z/2 | X/2 |
| Z | X/2, Y/2, X/2 |
| Z/2 | − Y/2, X |
| Z/2, X | X/2, Y |
| Z/2, X/2 | X/2, − Y/2, X/2 |
| Z/2, X/2, − Z | Y/2 |
| Z/2, X/2, − Z/2 | − X/2, Y |
| Z/2, X/2, Z/2 | X/2, Y/2, − X/2 |

**Table 2 | Clifford gate physical gate fractions for resonantly driven and dressed ST qubits**

| Resonant Gate Comp. | | Dressed Gate Comp. | |
|---|---|---|---|
| **Gate Type** | **Percentage** | **Gate Type** | **Percentage** |
| ± Z/2, ± Z | 58.82% | ± Y/2 | 35.6% |
| ± X/2 | 31.37% | ± Y | 8.9% |
| ± X | 7.84% | ± X/2 | 44.4% |
| I | 1.96% | ± X | 8.9% |
| | | I | 2.2% |

For the resonant qubit, there are 2.125 gates per Clifford on average and a 0.392 physical gate fraction; for the dressed qubit, 1.875 gates per Clifford and approximately 0.98 physical gate fraction.

arbitrary waveform generators (AWGs). Since the relevant frequencies in our experiments are below 200 MHz, the qubit drive pulses are directly synthesized by the AWG. The waveforms corresponding to P1, P2, and B12 were generated by the AWG channels 1, 2, and 4, respectively, while channel 3 was connected to the plunger gate of the SHT.

The SHT current is measured using a pair of Basel Precision Instruments (BasPI) SP983c IV converters with a gain of $10^6$, combined with a low-pass output filter with a cutoff frequency of 30 kHz. A source-drain bias excitation of $V_{sd} = 0.25$ mV is applied during the measurement. The differential current is extracted using a BasPI SP1004 differential amplifier with a gain of $10^2$, and the resulting signal is recorded with an Alazar ATS9440 digitizer card. The resulting time-resolved current was averaged over hundreds of experimental shots.

### Randomized Benchmarking
**Resonantly driven ST qubit.** The Clifford gates for single-qubit RB are composed as per Table 1[9,41]. The average number of gates per Clifford

is 2.125, and the percentage of physical gates is 0.392 (see Table 2). Average physical gate infidelity is calculated as $(1 - F_C)/2.125/0.392$, where $F_C$ is the average Clifford gate fidelity. The provided $1\sigma$ uncertainty is computed with respect to the nonlinear least squares fit as the square root of the corresponding diagonal element of a covariance matrix (using scipy.curvefit).

**Dressed ST qubit.** The Clifford gates for single-qubit RB are composed as per Table 1[9,41]. The average number of gates per Clifford is 1.875, and the percentage of physical gates is approximately 0.98 (see Table 2). The average physical gate infidelity is calculated as $(1 - F_C)/1.875/0.98$.

## Data availability
The data and analysis that support the findings of this study are available in a Zenodo repository https://doi.org/10.5281/zenodo.17144607. Supplementary information is available with this paper.

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

## Acknowledgements

This project acknowledges funding from the European Union's Horizon 2020 research and innovation program under the Marie Skłodowska-Curie grant agreement no. 847471. In addition, this project has received funding from the NCCR SPIN under grant no. 51NF40-180604 and 51NF40-225153, and under the grant no. 200021-188752 of the Swiss National Science Foundation. We also thank Michael Stiefel and all the Cleanroom Operations Team of the Binnig and Rohrer Nanotechnology Center (BRNC) for their help and support.

## Author contributions

K.T. and U.v.L. conceived the experiment. K.T. performed the experiment with help from U.v.L. and A.O. B.H. and K.T. simulated the data with the help of G.S. A.O. and U.v.L. took and evaluated the randomized benchmarking data with help from B.H. F.J.S. and M.M. fabricated the device. K.T. and N.W.H. designed the device mask based on previous work by L.M., L.S., I.S., K.T., F.J.S., P.H.C., and M.M. developed parts of the device. S.B. grew the heterostructure. G.S., I.S., P.H.C., and N.W.H. developed the measurement software. S.P., M.A., E.G.K., and G.S. contributed to the development of the experimental setup. M.P.V. contributed to discussions and provided feedback to the manuscript. K.T., U.v.L., A.O., B.H., and P.H.C. wrote the manuscript with input from all authors. A.F. and P.H.C. supervised the project.

## Competing interests

The authors declare no competing interests.
