## [Transparent Peer Review file · Nature Communications]

A dressed singlet-triplet qubit in germanium

Corresponding Author: Dr Patrick Harvey-Collard

Version 0:

Reviewer comments:

Reviewer #1

(Remarks to the Author)

In this work, the authors have implemented a dressed ST qubit on a germanium device. Compared to conventional single-spin qubits, dressed ST qubits offer several advantages, including lower required microwave frequencies, reduced crosstalk during manipulation, and longer coherence times (T_2). Additionally, the fidelity of both the ST qubit and the dressed ST qubit was characterized through randomized benchmarking (RB), exceeding 99.6%. This approach opens the door to more efficient operations in semiconductor-based quantum processors. However, it is somewhat concerning that while the authors claim the dressed ST qubit demonstrates greater resilience to charge noise, a direct improvement in fidelity was not observed. Although there are still some details that need to be perfected, I believe their work has a positive significance for subsequent development. I recommend its publication after the authors have addressed the following points in manuscript

1. In this device, the parameters are approximately $dE_z \approx 16$ MHz and $J \approx 6$ MHz. Could you clarify why the JAC component in the σ_z direction can be neglected?
2. Why is the duration for the $X\pi$ gate not simply double that of the $X\pi/2$ gate?
3. The gate sets utilized in the two randomized benchmarking (RB) experiments differ, with one of the methodologies incorporating virtual gates. Can the fidelities from these two setups be directly compared in this manner? I suggest that the authors provide a more detailed explanation for the lack of improvement in fidelity.
4. The dressed ST qubit involves two instances of the rotating wave approximation (RWA) with parameters $F=16$ MHz, $f=4.2$ MHz and $F=4.2$ MHz, $f=1$ MHz. What is the impact of these two approximations on the final fidelity? Has it been quantified?
5. Regarding Figure.S1(c) in the supplementary materials, I do not understand why the experimental results can characterize the difference between S_{02} and T_- .
6. What do you think is the core issue limiting fidelity? Is it due to charge noise? Does the waveform added to J not being a strict sine form have an impact? Has the error magnitude it introduces been subjected to rigorous theoretical calculation?
7. Authors mention that ST qubit can be realized in low magnetic field. However, low magnetic field result in a small Zeeman energy difference. According to section D of the supplementary information, if the Zeeman energy difference is small, it will lead to a low gate speed. Furthermore, if Zeeman energy difference is too small, maybe it is impossible to define ST qubit. I would like to hear your understanding.
8. In Section VI, authors say that the driving amplitude of the dressed ST qubit can be increased by increasing the amplitude of frequency modulation rather than using a higher voltage which could cause heating and crosstalk, and it is limited only by the capabilities of the control electronics. However, the driving amplitude is also limited by the two successive rotating wave approximations. In fact, based on the setup in this work ($f_{ST}=15.65$ MHz and $\Omega_{ST}=4.2$ MHz), the gate speed could not be particularly fast compared to the single hole spin qubit in low magnetic field (Nat. Mater. 23, 920–927 (2024)) even increasing the amplitude of frequency modulation. Could you provide further clarification?
9. The dressed ST qubit in this work exhibits a shorter T_1 compared to the resonant ST qubit, and the fidelity shows no significant improvement. Although the dressed ST qubit exhibits a longer T_2 , more advanced dynamic decoupling

sequences and better quantum devices could further extend T2 of resonant ST qubit, in fact, the upper limit of T2 is fundamentally constrained by T1. Given this, what do you think the advantage of the dressed ST qubit over the resonant ST qubit? Furthermore, what advantages do both types of ST qubit offer when compared to the single hole spin qubit?

10. In section A of the supplementary information, the ramp time of initialization and readout are investigated very clearly. However, based on Figure S2c, SPAM fidelity still does not reach 99%. In section B of the supplementary information, the SPAM error after X gate is explained, but the SPAM error after identity operation is not mentioned, what factors do you consider to be the primary limitations of the SPAM fidelity after identity operation?

Reviewer #2

(Remarks to the Author)

The manuscript "A dressed singlet-triplet qubit in germanium" by K. Tsoukalas et al., brings together three timely topics of semiconductor quantum dot spin qubits: i) Hole spin qubits in Ge/SiGe heterostructures, ii) qubit encoding in the singlet-triplet subspace (in this case in the anti-polarized states) of a double-dot, with qubit control implemented via modulation of the exchange interaction, and iii) continuous driving to encode the qubits in the dressed singlet-triplet states to achieve longer coherence times. While each of these topics on its own is nothing new, the combination of all three makes for an interesting and novel implementation. Furthermore, the manuscript is well written, and the data is of high quality.

We do have a few questions and comments that we would like to see addressed before we can recommend publication:

I. Introduction:

I.a. The third paragraph of the introduction states that ST qubits do not have non-orthogonal single qubit control axes. However, the paper Burkard et al. 2023 Review of Modern Physics clearly shows two-orthogonal axes for S-T0 qubits, with the inclusion of DC magnetic field gradient. Could you elaborate more on the non-orthogonality of these axes?

I.b. The last paragraph of the introduction compares the gate fidelities and coherence times of the bare and the dressed qubits. Another important metric is also the time it takes to perform the gates, which should be stated here for completion.

II. Device and Measurement Protocol:

II.a. The manuscript misses a section (either in the methods or the supplementary material) that describes the experimental setup. What kind of equipment (cryostat, electronics, amplifiers, etc.) was used and how the measurements were implemented.

II.b. The two quantum dot gates are referred to as "virtual gates vP1 and vP2". What makes these gates "virtual"? In the community the term is used to describe gates that are compensated for cross-capacitances with neighbouring gates, however the charge stability map in Fig. 1b clearly shows that there is a cross-capacitive coupling between vP1 and vP2.

II.c. The caption for the SEM image in Fig. 1a describes the image being "of the six QD device used in the experiments". Was the SEM image taken after the measurement campaign was completed, or is the deposition of negative charges from the electron beam not an issue for Ge/SiGe devices?

II.d. According to the caption of Fig. 1b, the detuning ϵ is defined "in units of its vP1 coordinate", which implies a voltage unit. The text, however, states " $\epsilon = 10$ meV", which would be an energy unit.

II.e. Do we interpret the manuscript correctly that all measurements are time-averaged measurements of the sensor current? At least, the caption for Fig. 1d states that "P is calibrated using the average current values in the (2,1) and (1,1) regions.", and similar statements can be found elsewhere. What is the actual value of the sensor current and by how much does it change during readout? The use of "arbitrary units" everywhere in the paper is a bit unsatisfactory here. At least Fig. 1b,c, Fig. 3a,c, Fig. S1a,c,d,e,f, Fig. S2b, and most importantly Fig. S2c could be plotted in units of pA or nA.

II.f. The previous point implies that the authors have NOT performed single-shot readout for any of the data. Why was this not done? Fig. S2c shows very good charge readout fidelity with two clearly separated histogram peaks and the last paragraph of the Supplementary Section B states that Fig. S2c shows the histograms of 1000-shots measurements and "the device could allow for operation in the single-shot readout regime".

III. Resonantly-Driven ST Qubit:

III.a. There is a discrepancy between the direction of J_{AC} in the text and in the figure. In the text, J_{AC} is defined along σ_x , while in Fig. 1e it points along $-X$.

III.b. The fact that the applied, sinusoidal driving voltage gets translated into the exponential of a sinusoid, means that the system is not actually driven with a sinusoid as the authors point out in this section and in Fig. S3. Have the authors tried to compensate their driving waveform for this by applying a voltage proportional to $\log(\sin)$?

III.c. The implementation of randomized benchmarking states that the "Clifford gate set $\{X_{\pi}, X_{\pi/2}, Z_{\pi}, Z_{\pi/2}, I\}$ " is used. This definition is incorrect. The single qubit Clifford gate set consists of 24 different gates, and Clifford randomized benchmarking uses sequences that are randomly constructed out of these 24 gates. The paper by Epstein et al. is a good

reference for this: <http://dx.doi.org/10.1103/PhysRevA.89.062321> . We recommend deleting the word “Clifford” here and as appropriate in the whole paragraph.

The text also states that “ Z_{π} , $Z_{\pi/2}$, I are considered instantaneous and error-free” as they are only implemented virtually. We understand the purpose of a virtual Z gate, but a virtual I gate is not doing anything at all and should just be omitted from the gate set. If an identity is included in the gate set, then it should be implemented as a proper idle gate with a certain duration (e.g., duration of $X_{\pi/2}$).

III.d. Related to the previous comment, in the caption of Fig. S1a, the authors state that they apply an identity gate for the measurement in the left panel. Was this actually an identity gate or did they just do nothing?

V. Dressed ST Qubit Spectroscopy:

V.a. Where do the additional features next to the center peak in Fig. 3a come from? Are pump and probe coherent? The authors state in the caption for Fig. S4 that “The phase of the second tone is shifted by $\pi/4$ to get better agreement with the experimental data...”, which implies that there was a fixed relation between pump and probe beam. It would be interesting for the readers to see how pump and probe beam are generated, which relates back to our previous comment about the missing description of the experimental setup.

VI. Dressed ST Qubit Control:

VI.a. Fig. 3b-ii shows the dressed qubit Rabi frequency Ω_{ST}^{FM} as a function of the frequency modulation $\Delta\nu_{FM}/2$ up to 3 MHz. This is implemented for $\Omega_{ST} = 4.2$ MHz. For the fastest driving, the driving speed is 70% of the level splitting. We are a bit surprised that there are no effects of breakdown of the rotating wave approximation visible in the figure. Not even a fading of the oscillations towards higher driving speeds as described in Section E of the supplementary material. See Ref. 34 for comparison. Important here again is if there is a fixed phase relation between the RF sources.

VI.b. Related to the previous comment, Supp. Section E states that “When driving with $\Delta\nu_{FM} = 1$ MHz, we observe that the oscillations begin to fade at higher modulation frequencies (f_{FM}), indicating the breakdown of the rotating wave approximation.”

Could the authors please add some extra details on this discussion? We can see that in Fig. S5 the oscillations for higher f_{FM} are more faded than for lower f_{FM} , but the reason for this is not clear. The authors describe this as “indicating the breakdown of the rotating wave approximation”, but a higher f_{FM} does not correspond to a higher driving speed, and in fact, a lower f_{FM} (at constant driving speed) brings the counterrotating term closer to the transition frequency.

VI.c. The sentence “Additionally, in this scheme, increasing the drive amplitude does not necessitate a higher voltage, which could cause heating or crosstalk [35, 36], but rather a broader bandwidth, limited only by the capabilities of the control electronics [18].” can be made clearer, e.g. by inserting “a broader modulation bandwidth $\Delta\nu_{FM}$,...”.

VI. d. The randomized benchmarking of the dressed ST qubit is also done with a set of 5 qubit gates. This time the text states that “that all gates in the set are physical”. Does this include the identity gate?

Supplementary Section A:

S.a. The way this section reads makes it hard to understand which state is blockaded and which is not. In Fig. S1a (and the first paragraph of the section where this figure is explained), the down-down state seems to be unblockaded. On the other hand, in Fig S1d (and the third paragraph) states that the down-down state is blockaded. Then only Fig. S1e shows that the blockaded state is changing depending on the $t_{\text{ramp_out}}$, indicating that the $t_{\text{ramp_out}}$ in Fig. S1a must also be $1\mu\text{s}$. To minimize confusion, we suggest adding a pulse diagram to Fig. S1a and replace the sentence “(discussed in detail in the next paragraph)” with “(discussed in detail in the third paragraph)”.

S.b. In the second last paragraph of the Section A, only Fig. S1d is referred to while discussing the chirped pulse experiment. However, that panel does not involve any chirped pulse. Should the discussion refer to the Fig. S1e instead? Similarly, the last paragraph should refer to the Fig. S1f.

Reviewer #3

(Remarks to the Author)

The authors report the resonant driving of a singlet-triplet qubit in germanium by modulating the exchange coupling. Furthermore, by continuously driving the qubit they demonstrate the dressed singlet-triplet qubit with an increased T_2^* of 20.3 μs . The reported single qubit gate fidelity of 99.68% is among the highest reported values in literature. Overall, the paper is well-written, the storyline is clear, and the measurement data is of high quality. I will be happy to support its publication in Nature Communications after the authors help me clarify a few points in the manuscript.

1)The dressed qubit shows a ten-fold increase in T_2^* compared to the bare qubit. However, the gate fidelity is not higher. In that case, does the dressed qubit have advantages over the bare qubit?

2)What are the limiting factors for gate fidelities of the bare qubit and the dressed qubit, respectively?

3) In fig 3(c), the authors use normalized I_{sensor} to fit T_1 and T_2 . The measurement noise seems quite high compared to other data. Why do they plot fig 3(c) using I_{sensor} instead of the state population as in other figures?

4) When characterizing the dressed qubit fidelity the authors use a gate set which is different from the set they use for the bare singlet-triplet qubit. What is the reason behind this choice? I suppose using the same gate set would make better one-to-one comparison?

Reviewer #4

(Remarks to the Author)

Version 1:

Reviewer comments:

Reviewer #1

(Remarks to the Author)

After reviewing the author's responses to the referees, I have no further concerns on the manuscript and recommend its acceptance in its current form for Nature Communications.

Reviewer #2

(Remarks to the Author)

We thank the authors for taking the time to diligently answer our questions and address our comments. We feel that not only our comments, but also those of the other reviewers have been satisfactorily addressed, and we support publication of this manuscript in Nature Communications.

When reading through the resubmitted version, we noticed 3 typos:

- Page 6, "but rather a broader bandwidth by inserting a broader modulation bandwidth $\Delta\nu_{\text{FM}}$ ": The words "by inserting" seem out of place here.
- Page 13, "Schödinger"
- Page 13, "Hamiltonian"

Reviewer #3

(Remarks to the Author)

The authors have answered my questions very well and revised the manuscript. Now the paper is more clear. Here I support its publication in Nature Communications.

Reviewer #4

(Remarks to the Author)

Dear Editors and Referees,

Thank you for your feedback about our manuscript, “A dressed singlet-triplet qubit in germanium”. We also want to thank the referees for taking the time to review our work.

Please find below our point-by-point response to the referees. We also provide a revised version of the manuscript with the changes marked using the latexdiff tool.

Sincerely,

K. Tsoukalas, A. Orekhov and P. Harvey-Collard, on behalf of all authors

REVIEWER COMMENTS

Reviewer #1 (Remarks to the Author):

In this work, the authors have implemented a dressed ST qubit on a germanium device. Compared to conventional single-spin qubits, dressed ST qubits offer several advantages, including lower required microwave frequencies, reduced crosstalk during manipulation, and longer coherence times (T₂). Additionally, the fidelity of both the ST qubit and the dressed ST qubit was characterized through randomized benchmarking (RB), exceeding 99.6%. This approach opens the door to more efficient operations in semiconductor-based quantum processors. However, it is somewhat concerning that while the authors claim the dressed ST qubit demonstrates greater resilience to charge noise, a direct improvement in fidelity was not observed. Although there are still some details that need to be perfected, I believe their work has a positive significance for subsequent development. I recommend its publication after the authors have addressed the following points in manuscript

1. In this device, the parameters are approximately $dE_z \approx 16$ MHz and $J \approx 6$ MHz. Could you clarify why the JAC component in the σ_z direction can be neglected?

If we understand the question correctly, the short answer is that it is a “longitudinal” term, and therefore cannot drive rotations. This topic is discussed in Eq. S6 in Sec. D of the Supplementary Information. There, we perform the rotating wave approximation (RWA), which results in the Hamiltonian:

$$H_{RWA}/\hbar = (f_{ST} - f_d)/2 \sigma_z + A_J/4 \sigma_x + A_J \cos(2\omega_d t)/2 \sigma_x - A_J \sin(2\omega_d t)/2 \sigma_y + A_{Jz} \cos(\omega_d t) \sigma_z$$

Here we clarify that:

$$A_J \sim \Delta E_z / f_{ST}$$

$$A_{Jz} \sim J_{DC} / f_{ST}$$

The JAC component along σ_z is neglected because, under the rotating wave approximation (RWA), it rotates rapidly at the drive frequency ω_d (see Eq. S6 in Sec. D of the Supplementary Information). This is analogous to the standard RWA argument used to neglect fast-oscillating terms (with frequency $2\omega_d$) along σ_x and σ_y . Moreover, the amplitude of the JAC σ_z term (A_{Jz}) is approximately three times smaller than that of the fast-oscillating transverse components (A_J), making its contribution even less significant. See also the response to question 6.

We have clarified this in the SI text in the following:

in the lab frame where $\sigma_z = |\downarrow\uparrow\rangle\langle\downarrow\uparrow| - |\uparrow\downarrow\rangle\langle\uparrow\downarrow|$. We note that a sinusoidal drive on the barrier gate would induce an oscillating exchange of the form $\exp(A\sin(\omega t))$. However, since the $e^{A\cos\omega t} \approx J_0(A) + 2J_1(A)\cos(\omega t) + 2J_2(A)\cos(2\omega t) + \dots$, where $J_n(A)$ is the modified Bessel function of the first kind of order n evaluated at A . Since the upper harmonics in the Fourier series are far off-resonant, in the following and have a lower amplitude than the first one, we lump the zeroth Fourier component $J_0(A)$ into J_{DC} and consider only the first component $\propto \sin(\omega t) \propto \cos(\omega t)$ for the drive. At a higher order, $J_{DC} \rightarrow J_{DC}(A)$ leads to a change in the idling versus driven qubit frequency that is smaller than the qubit linewidth, and therefore we neglect it. Furthermore, during all RB experiments, the waveforms are assembled without any gaps between the gates, so this effect is avoided altogether. Alternatively, one could in principle “easily” correct for this by tracking the qubit phase shift and applying phase corrections.

Defining the exchange drive along σ_x as $A_J = J'_{AC}\Delta E_z/hf_{ST}$ and omitting the second term in the driven part of the exchange drive along σ_z as $A_{Jz} = J'_{AC}J_{DC}/hf_{ST}$, we arrive at Eq. (S1) of the main text, i.e.,

$$H_{res}^{ST}/h = \frac{f_{ST}}{2}\sigma_z + \frac{J_{AC}(t)}{2}\sigma_x + \frac{J_{ACz}(t)}{2}\sigma_z, \quad (S3)$$

where $J_{AC}(t) = A_J \cos(\omega_d t)$, $J_{AC}(t) = A_J \cos(\omega_d t)$ and $J_{ACz}(t) = A_{Jz} \cos(\omega_d t)$ with $\omega_d = 2\pi f_d$. In the following we note that for the values of $J_{DC} \approx 6$ MHz and $\Delta E_z/h \approx 16$ MHz we get $A_J \approx 3A_{Jz}$. Below, we will see that the omitted term is indeed negligible in the rotating-wave approximation (RWA). $J_{ACz}(t)\sigma_z$ term can be omitted and therefore we arrive at Eq. (S1) of the main text.

Rotating frame. The time-dependent Schrödinger equation with the Hamiltonian of Eq. (S2) can be solved in the

2. Why is the duration for the $X\pi$ gate not simply double that of the $X\pi/2$ gate?

This is a commonly observed behavior in spin qubit gates and can arise from several sources:

- Drive-related effects: Slight detuning from the qubit resonance, as well as drive-induced shifts such as AC Stark shifts may accumulate non-linearly with pulse duration.
- Pulse shaping effects: Non-instantaneous rise times and waveform distortions arising from nonlinearities in the control electronics can result in non-ideal pulse envelopes.

We will clarify this in the text:

and $X_{\pi/2}$ durations, corresponding to oscillation speeds of 0.5 and 0.25, respectively. The fact that the duration of the $X_{\pi/2}$ gate is not exactly half that of the X_{π} gate may originate from various sources, including non-ideal pulse shaping (e.g., finite rise times) and drive-duration-dependent effects such as AC Stark shifts or local heating.

3. The gate sets utilized in the two randomized benchmarking (RB) experiments differ, with one of the methodologies incorporating virtual gates. Can the fidelities from these two setups be directly compared in this manner?

Indeed, the gate sets used in the two experiments differ, with the resonantly driven one containing (virtual) Z gates while the dressed ST contains physical Y gates.

As is usual, we only accounted for the physical gates performed on each experiment to calculate the stated average physical gate fidelity. The assumption that virtual gates are error free simply shifts the error budget to the physical gates.

For the resonant ST qubit this amounts to: 51/24 gates per Clifford and 0.392 physical gates per gate.

For the dressed ST qubit this amounts to: 45/24 gates per Clifford and 0.98 physical gates per gate.

In addition, because the two different gate schemes have such fundamentally different drive mechanisms, we believe that even an identical RB sequence would not offer any advantage in terms of comparing them.

We note that we have made 2 corrections from the previous version, based on other reviewer comments (see below), that do not influence significantly the fidelities

- 1) Regarding the I gate in the dressed scheme. It was mistakenly counted as physical although it was a do-nothing-for-no-time gate. This has negligible impact in the fidelity, changing its central value from 99.64(2) to 99.63(2).
- 2) Regarding the average number of gates per Clifford for the resonant ST qubit, we have corrected the previously stated 2.134 to 2.125. The mistake occurred because we had made this calculation from an experimental sequence. This affects the fidelity below the error margin and hence no correction is needed.

We have added a new section describing our RB analysis in detail, please see the reply to Reviewer 3, question IIIc.

I suggest that the authors provide a more detailed explanation for the lack of improvement in fidelity.

We assume that the expectation for a fidelity improvement stems from comparing the T_2^* and t_{gate} and since T_2^* increased by an order of magnitude, more than the slowdown of the gate, and therefore the fidelity should increase.

We would first like to note that another relevant timescale here would be the T_2^{Rabi} which was not measured for the dressed qubit. Assuming that T_2^{Rabi} would have increased similarly to T_2^* is not straight forward because the different nature of the dressed qubit driving (FM) mechanism can result to a very different noise sampling. For example, the T1 is shorter. An improved dressing protocol that filters noise more effectively has been developed to address this question (The SMART protocol).

In addition, as the reviewer also suggests in the next question, the FM driven dressed qubit involves 2 RWA which have both a Larmor/Rabi frequency ratio ≈ 0.25 , closer to the RWA limit than the resonant ST. This was done to accelerate the gate and reduce decoherence errors but will narrow the parameter space for fine tuning the gates, hence opening up room for unitary or calibration errors.

Due to the complexity of the error mechanisms that can influence qubit gates to different kind of qubits, we regarded it necessary to perform RB characterization and not only assume the improvement due to the longer coherence.

Furthermore, the fidelity optimization will be left for future work.

We have added these comments in the main text as:

the error margin compared with the bare S_1 qubit. Dressing the qubit has improved fidelity in some cases (e.g., Vallabhapurapu *et al.* [39]) but not in others (e.g., Hansen *et al.* [20]). The lack of fidelity improvement in our work could be related to both unitary and decoherence errors. Unitary errors could be introduced by the high drive speed used to reduce the gate duration, challenging the RWA. A reduction of decoherence errors can be achieved by an additional improvement of the noise filtering through the application of more elaborate dressing schemes [20] and reduced nuclear spin noise [9]. Even without improvement in the gate fidelity, the advantage of the dressed scheme lies in the extended coherence time, which is especially beneficial during idle periods such as readout or two-qubit gate operations elsewhere in the device. Compared with CPMG, the dressing approach offers better synchronicity and is simple to implement. In the future, the drive signal on the barrier could be transformed to compensate for the exponential dependence of the exchange interaction on the voltage, suppressing the higher order terms [40].

These findings demonstrate, on the one hand, the highly

4. The dressed ST qubit involves two instances of the rotating wave approximation (RWA) with parameters $F=16\text{MHz}$, $f=4.2\text{MHz}$ and $F=4.2\text{MHz}$, $f=1\text{MHz}$. What is the impact of these two approximations on the fidelity? Has it been quantified?

While the impact of the two approximations on the fidelity has not been quantified theoretically, we have used accepted benchmarking techniques to demonstrate their real-life performance. We can see that the ratio of the magnitude of the drive frequencies (~ 0.25 Larmor freq) approaches the upper limit of validity for the RWA. However, it is still possible to achieve high gate fidelities with appropriate parameter choices due to a synchronization. In practice, the rotation angle calibration is experimentally done, and both the drive frequency (which can shift at large drive amplitudes) and the timing of the $\pi/2$ and π gates were calibrated independently.

However, indeed by operating in this regime a more constrained parameter space for fine tuning the gate exists. We can understand that this can affect the stability of the gate and may ultimately limit the achievable fidelity due to miscalibrations.

We would also like to mention that recent developments in the field of strongly driven qubits have introduced protocols that mitigate errors arising from the breakdown of the RWA, offering promising paths forward in this regime. (Zwanenburg *et al.* Single-Qubit Gates Beyond the Rotating-Wave Approximation for Strongly Anharmonic Low-Frequency, <https://arxiv.org/abs/2503.08238>)

Regarding the changes to the manuscript, please see the added section of the previous question.

5. Regarding Figure.S1(c) in the supplementary materials, I do not understand why the experimental results can characterize the difference between S02 and T-

If we correctly understand the question, it is a standard technique that adiabatically controls the mapping of states between the control region and the readout region, similar Zhang et al. Nat. Nanotechnol. 20, 209–215 (2025) and E. G. Kelly et al. arXiv preprint arXiv:2504.06898 (2025). The measurement can be explained by the energy diagram of Fig. S1(b). The new preprint by E. G. Kelly et al. explains the readout principles in extensive details.

6. What do you think is the core issue limiting fidelity? Is it due to charge noise?

Aside from the likely unitary errors, we do not know quantitatively the impact of charge noise and nuclear spin noise on the fidelity. Some qualitative information regarding the noise source in the experiment can be extracted out of the refocusing pulses (T2* vs T2ECHO, etc). While these pulses show an increase in coherence time, this is moderate which indicates that there is noise spread over a large frequency spectrum and not only in low frequencies. Typically devices with very low charge noise show greater improvements with an echo sequence such as the one in Steinacker et al. <https://arxiv.org/abs/2410.15590>. However nuclear spin noise is likely also present as this is a natural isotope device and the field direction is not intentionally aligned with the nuclear noise suppression [Hendricks, Nat. Materials, 23, 920 (2024)].

Does the waveform added to J not being a strict sine form have an impact? Has the error magnitude it introduces been subjected to rigorous theoretical calculation?

Interesting point. This is already addressed partially and indirectly in Sup. Sec. C. RESONANT EXCHANGE INTERACTION, see the numerical QuTiP simulation, where we show that the exponentiated drive leads to a non-linear Rabi frequency. However we now add the following clarification in the SI as well. The Fourier series expansion of the function

$$e^{A \cos(\omega t)} \approx I_0(A) + 2I_1(A)\cos(\omega t) + 2I_2(A)\cos(2\omega t) + 2I_3(A)\cos(3\omega t) \dots$$

where $I_n(A)$ is the modified Bessel function of the first kind of order n evaluated at A .

We neglect the higher harmonic terms for the same reasoning used to neglect the counter-rotating terms in the Rotating Wave Approximation. Furthermore, these higher harmonics have smaller amplitudes compared to the first harmonic because $I_1(A) > I_2(A) > I_3(A)$, consistent with the known properties of the modified Bessel functions of the first kind. Since they are off-resonant, their contribution to driving is suppressed. Furthermore, since they are periodic and at integer multiples, and given that the gate times are calibrated experimentally, is possible that their error is even smaller than a naive off-resonant drive estimate would suggest. However we did not investigate this from a rigorous theoretical point of view.

The rectifying behavior of J has effects on the DC phase accumulation of single qubits. This is studied theoretically in other works, for instance [M. Rimbach-Russ, S. G. J. Philips, X. Xue, and L. M. K. Vandersypen, Quantum Science and Technology 8, 045025 (2023)].

We add

in the lab frame where $\sigma_z = |\downarrow\uparrow\rangle\langle\downarrow\uparrow| - |\uparrow\downarrow\rangle\langle\uparrow\downarrow|$. We note that a sinusoidal drive on the barrier gate would induce an oscillating exchange of the form $\exp(A \sin(\omega t))$. However, since the $e^{A \cos \omega t} \approx I_0(A) + 2I_1(A) \cos(\omega t) + 2I_2(A) \cos(2\omega t) + \dots$, where $I_n(A)$ is the modified Bessel function of the first kind of order n evaluated at A . Since the upper harmonics in the Fourier series are far off-resonant, in the following and have a lower amplitude than the first one, we lump the zeroth Fourier component $I_0(A)$ into J_{DC} and consider only the first component $\propto \sin(\omega t) \propto \cos(\omega t)$ for the drive. At a higher order, $J_{DC} \rightarrow J_{DC}(A)$ leads to a change in the idling versus driven qubit frequency that is smaller than the qubit linewidth, and therefore we neglect it. Furthermore, during all RB experiments, the waveforms are assembled without any gaps between the gates, so this effect is avoided altogether. Alternatively, one could in principle “easily” correct for this by tracking the qubit phase shift and applying phase corrections.

Defining the exchange drive along σ_x as $A_J = J_{AC} \Delta E_z / \hbar f_{ST}$ and omitting the second term in the driven part of the exchange drive along σ_z as $A_{Jz} = J_{AC} J_{DC} / \hbar f_{ST}$, we arrive at Eq. (S1) of the main text, i.e.,

$$H_{res}^{ST} / \hbar = \frac{f_{ST}}{2} \sigma_z + \frac{J_{AC}(t)}{2} \sigma_x + \frac{J_{ACz}(t)}{2} \sigma_z, \quad (S3)$$

where $J_{AC}(t) = A_J \cos(\omega_d t)$, $J_{AC}(t) = A_J \cos(\omega_d t)$ and $J_{ACz}(t) = A_{Jz} \cos(\omega_d t)$ with $\omega_d = 2\pi f_d$. In the following we note that for the values of $J_{DC} \approx 6$ MHz and $\Delta E_z / \hbar \approx 16$ MHz we get $A_J \approx 3 A_{Jz}$. Below, we will see that the omitted term is indeed negligible in the rotating-wave approximation (RWA), $J_{ACz}(t) \sigma_z$ term can be omitted and therefore we arrive at Eq. (S1) of the main text.

Rotating frame. The time-dependent Schrödinger equation with the Hamiltonian of Eq. (S2) can be solved in the

In writing this, we realized that there is a subtle but important behavior worth pointing out. The term $I_0(A) \neq 0$ leads to a small 2nd order frequency shift of order ~ 0.2 MHz (estimated from device parameters) of the qubit when it is driven vs when it is idling (i.e., due to the change in J_{DC} , eq. S2). This is smaller than the qubit's linewidth, therefore we do not worry too much about it. However, we avoid this effect altogether because we generate the pulses without any gaps between them: the qubit is always driven during RB. However, if the qubit idles, it would accumulate a phase with respect to the driving clock. This is not a fundamental problem, as one can anticipate this and compensate it with virtual Z rotations, but it is nevertheless good to mention.

7. Authors mention that ST qubit can be realized in low magnetic field. However, low magnetic field result in a small Zeeman energy difference. According to section D of the supplementary information, if the Zeeman energy difference is small, it will lead to a low gate speed. Furthermore, if Zeeman energy difference is too small, maybe it is impossible to define ST qubit. I would like to hear your understanding.

All of these points are valid. The motivation for reducing the magnetic field in hole spin qubits is to increase coherence time enough, so that instead of being limited by charge noise via spin-orbit coupling, it becomes nuclear-spin limited. Beyond this point, there is no further T_2^* improvement. There is indeed a tradeoff between coherence and gate speed in resonant gates. However, this holds only above a certain threshold (~ 5 mT Ref. Wang et al. Science 385, 447 (2024)). To make a fundamental improvement outside of this trade off, the magnetic field direction should be optimized as done in Hendrickx et al. Nat. Mater. 23, 920–927 (2024) with a vector magnet.

Add to paper:

ST systems [11, 12, 33]. Although operation at low B increases the dephasing time, we note that there is a trade-off between dephasing time and gate speed (for resonant gates) which can be fundamentally improved by operating in specific B field directions [9]. A Rabi

8. In Section VI, authors say that the driving amplitude of the dressed ST qubit can be increased by increasing the amplitude of frequency modulation rather than using a higher voltage which could cause heating and crosstalk, and it is limited only by the capabilities of the control electronics. However, the driving amplitude is also limited by the two successive rotating wave approximations. In fact, based on the setup in this work ($f_{ST}=15.65$ MHz and $\Omega_{ST}=4.2$ MHz), the gate speed could not be particularly fast compared to the single hole spin qubit in low magnetic field Nat. Mater. 23, 920–927 (2024) even increasing the amplitude of frequency modulation. Could you provide further clarification?

Again all of these points are valid.

For the dressed ST qubit in our work compared to the Nat. Mater. case, both speed and T_2^* are indeed similar. However, in that work, the magnetic field orientation was optimized to operate at a hyperfine sweet spot (very slightly out of plane), and a balance between low charge noise sensitivity and high operation speed is achieved.

Applying a dressing scheme under sweet spot conditions for our ST qubit could yield significantly better performance, this is a promising direction for future research. Unfortunately we did not have a vector magnet in this set up.

We note that the driving mechanisms of the two qubits in the two experiments is completely different. In the case of the Nat. Mat., the drive of the single spin is done by G TMR with an optimized B that maximizes your drive speed. However, we drive the ST by the exchange interaction which for example does not require a large G TMR so the gate speed is ultimately constrained by very different things, for the better and the worse.

9. The dressed ST qubit in this work exhibits a shorter T1 compared to the resonant ST qubit, and the fidelity shows no significant improvement. Although the dressed ST qubit exhibits a longer T2, more advanced dynamic decoupling sequences and better quantum devices could further extend T2 of resonant ST qubit, in fact, the upper limit of T2 is fundamentally constrained by T1.

Given this, what do you think the advantage of the dressed ST qubit over the resonant ST qubit?

Furthermore, what advantages do both types of ST qubit offer when compared to the single hole spin qubit?

Valid points.

The advantage of the dressed scheme lies in the extended T_2 , which is especially beneficial during idle periods such as readout or two-qubit gate operations elsewhere in the device.

We would like to remark that, while T_2 is upper bounded by 2^*T_1 , depending on the noise spectrum and the gate speed it is not always possible to reach this limit; therefore one cannot assume that the resonant ST qubit CPMG performance would reach 2^*T_1 , in fact we are convinced it would not. Characterizing real-life gate fidelity with rigorous benchmarking methods can help avoid some of these loopholes.

Compared to CPMG, the dressing approach offers natural synchronicity with the single qubit gates. CPMG is discrete in time, while dressing shifts the reference frame continuously. This makes single-qubit operations straightforward under dressing, whereas with CPMG, gate timing must avoid overlap with refocusing pulses, though we agree that it is not clear if this is really a big problem.

We add

noise [9]. Even without improvement in the gate fidelity, the advantage of the dressed scheme lies in the extended coherence time T_2^* , which is especially beneficial during idle periods such as readout or two-qubit gate operations elsewhere in the device. Compared with CPMG, the dressing approach offers better synchronicity and is simple to implement. Additionally, the drive signal on the barrier could be transformed to compensate for the exponential dependence of the exchange interaction on the voltage, suppressing the higher order terms [39].

We would like to make clear that we do not claim that the ST qubit are clearly superior to the single spins, nor the other way around. However, there are certain clearly appealing characteristics of ST vs single-hole spin qubit (LD):

1. Faster drive due to highly tunable J which is more engineerable than the G tensor modulation in germanium for larger systems
2. No additional ancilla for readout is needed, it is built into the encoding. (LD qubits need to drag ancillas or suffer serial readout)
3. ST Larmor frequency $\sim \Delta E_z$, which can provide resilience to certain correlated noise sources such as global magnetic field fluctuations

10. In section A of the supplementary information, the ramp time of initialization and readout are investigated very clearly. However, based on Figure S2c, SPAM fidelity still does not reach 99%. In section B of the supplementary information, the SPAM error after X gate is explained, but the SPAM error after identity operation is not mentioned, what factors do you consider to be the primary limitations of the SPAM fidelity after identity operation?

This is a very important question but in this work we did not focus on the origin or the exact magnitude of the SPAM errors, as we performed randomized benchmarking which is unaffected by SPAM (in principle...). We made a separate study about readout and SPAM in a different device which is now available as a preprint: E. G. Kelly et al. arXiv preprint arXiv:2504.06898 (2025).

We add:

window. A more detailed analysis of the magnitude of SPAM errors and the optimization of readout mechanisms is presented in Kelly *et al.* [28].

Reviewer #2 (Remarks to the Author):

The manuscript “A dressed singlet-triplet qubit in germanium” by K. Tsoukalas et al., brings together three timely topics of semiconductor quantum dot spin qubits: i) Hole spin qubits in Ge/SiGe heterostructures, ii) qubit encoding in the singlet-triplet subspace (in this case in the anti-polarized states) of a double-dot, with qubit control implemented via modulation of the exchange interaction, and iii) continuous driving to encode the qubits in the dressed singlet-triplet states to achieve longer coherence times. While each of these topics on its own is nothing new, the combination of all three makes for an interesting and novel implementation. Furthermore, the manuscript is well written, and the data is of high quality.

We do have a few questions and comments that we would like to see addressed before we can recommend publication:

I. Introduction:

I.a. The third paragraph of the introduction states that ST qubits do not have non-orthogonal single qubit control axes. However, the paper Burkard et al. 2023 Review of Modern Physics clearly shows two-orthogonal axes for S-T0 qubits, with the inclusion of DC magnetic field gradient. Could you elaborate more on the non-orthogonality of these axes?

The point we aimed to make is that, in the conventional S-T₀ qubit, the magnetic field gradient ΔB_z is typically fixed and cannot be turned off; meanwhile the exchange interaction $J(\epsilon)$ is electrically tunable but not so large that ΔB_z can be completely neglected (especially when aiming for $F > 99\%$). Therefore, while $J(\epsilon)$ alone is orthogonal to ΔB_z , the qubit control axis always has a fixed component along ΔB_z , and one cannot rotate about orthogonal axes without using composite pulses.

We add:

non-orthogonal control axes arising from the combination of J and the non-tunable ΔE_z . Meanwhile, achieving

I.b. The last paragraph of the introduction compares the gate fidelities and coherence times of the bare and the dressed qubits. Another important metric is also the time it takes to perform the gates, which should be stated here for completion.

We agree and have now included the gate durations for both the bare and dressed qubits in the revised introduction.

with X_π gate durations of 327 ns and 500 ns

II. Device and Measurement Protocol:

II.a. The manuscript misses a section (either in the methods or the supplementary material) that describes the experimental setup. What kind of equipment (cryostat, electronics, amplifiers, etc.) was used and how the measurements were implemented.

We agree and we add the following section:

H. EXPERIMENTAL SETUP

All measurements are performed in a Bluefors XLD dilution refrigerator with a base temperature of 20 mK. The sample is mounted on a QDevil QBoard circuit board. Static gate voltages are applied via a QDevil QDAC through DC lines, which are filtered at the millikelvin stage using a QDevil QFilter. All plunger and barrier gates are connected to coaxial lines through on-PCB bias tees. RF lines are attenuated by 20 dB inside the refrigerator. Fast voltage excitation pulses are delivered to the quantum dot gates using Tektronix AWG5204 arbitrary waveform generators (AWGs). Since the relevant frequencies in our experiments are below 200 MHz, the qubit drive pulses are directly synthesized by the AWG. The waveforms corresponding to P1, P2 and B12 were generated by the AWG channels 1, 2, 4 respectively while the channel 3 was connected to the plunger gate of the SHT.

The SHT current is measured using a pair of Basel Precision Instruments (BasPI) SP983c IV converters with a gain of 10^6 , combined with a low-pass output filter with a cutoff frequency of 30 kHz. A source-drain bias excitation of $V_{sd} = 0.25$ mV is applied during the measurement. The differential current is extracted using a BasPI SP1004 differential amplifier with a gain of 10^2 , and the resulting signal is recorded with an Alazar ATS9440 digitizer card. The resulting time-resolved current was averaged over hundreds of experimental shots.

II.b. The two quantum dot gates are referred to as “virtual gates vP1 and vP2”. What makes these gates “virtual”? In the community the term is used to describe gates that are compensated for cross-capacitances with neighbouring gates, however the charge stability map in Fig. 1b clearly shows that there is a cross-capacitive coupling between vP1 and vP2.

We thank the reviewer for the observation. While we compensate for cross-capacitance between physical gates, some residual coupling between vP1 and vP2 remains in this regime, as evidenced by the stability diagram. We have now added the full virtual matrix used in the experiment in the SI:

the two virtual gates [26] vP1 and vP2 (Supplementary Information (SI) Sec. G), we measure the charge stability

Table S1. Virtual gate matrix.

	vP1	vP2	vP3	vP4	vP5	vP6	vB12	vB23	vB34	vB45	vB56	vBO1	vB6O	vBM	vS1P	vS1B1	vS1B2
P1	1.0	0	0	0	0	0	0	0	0	0	0	0	0	0	0	0	0
P2	0	1.0	0	0	0	0	0	0	0	0	0	0	0	0	0	0	0
P3	0	0	1.0	0	0	0	0	0	0	0	0	0	0	0	0	0	0
P4	0	0	0	1.0	0	0	0	0	0	0	0	0	0	0	0	0	0
P5	0	-0.015	-0.049	-0.14	1.0	-0.18	0	-0.03	-0.15	-0.575	-0.56	0	-0.05	-0.45	0	0	0
P6	0	0	-0.02	-0.075	-0.2	1.0	0	-0.013	-0.03	-0.075	-0.65	0	-0.14	-0.2	0	0	0
B12	0	0	0	0	0	0	1.0	0	0	0	0	0	0	0	0	0	0
B23	0	0	0	0	0	0	0	1.0	0	0	0	0	0	0	0	0	0
B34	0	0	0	0	0	0	0	0	1.0	0	0	0	0	0	0	0	0
B45	0	0	0	0	0	0	0	0	0	1.0	0	0	0	0	0	0	0
B56	0	0	0	0	0	0	0	0	0	0	1.0	0	0	0	0	0	0
BO1	0	0	0	0	0	0	0	0	0	0	0	1.0	0	0	0	0	0
B6O	0	0	0	0	0	0	0	0	0	0	0	0	1.0	0	0	0	0
BM	0	0	0	0	0	0	0	0	0	0	0	0	0	1.0	0	0	0
S1P	-0.005	-0.004	-0.01	-0.027	-0.044	-0.025	-0.013	-0.029	-0.028	-0.02	0	0	-0.003	-0.4	1.0	0	0
S1B1	0	0	0	0	0	0	0	0	0	0	0	0	0	0	0	1.0	0
S1B2	0	0	0	0	0	0	0	0	0	0	0	0	0	0	0	0	1.0

II.c. The caption for the SEM image in Fig. 1a describes the image being “of the six QD device used in the experiments”. Was the SEM image taken after the measurement campaign was completed, or is the deposition of negative charges from the electron beam not an issue for Ge/SiGe devices?

We confirm that the SEM image is from a ‘sister’ device fabricated on the same chip and located nearby. The actual device used in the experiment was not imaged to avoid possible damage. This has been clarified in the figure caption.

We add:

nominally identical six QD device

II.d. According to the caption of Fig. 1b, the detuning ϵ is defined “in units of its vP1 coordinate”, which implies a voltage unit. The text, however, states “ $\epsilon = 10$ meV”, which would be an energy unit.

We thank the reviewer for catching this. It was indeed a typo. The correct unit is voltage, and we have corrected this in the text.

II.e. Do we interpret the manuscript correctly that all measurements are time-averaged measurements of the sensor current? At least, the caption for Fig. 1d states that “ I_P is calibrated using the average current values in the (2,1) and (1,1) regions.”, and similar statements can be found elsewhere. What is the actual value of the sensor current and by how much does it change during readout? The use of “arbitrary units” everywhere in the paper is a bit unsatisfactory here. At least Fig. 1b,c, Fig. 3a,c, Fig. S1a,c,d,e,f, Fig. S2b, and most importantly Fig. S2c could be plotted in units of pA or nA.

The measurements are time-resolved and averaged over the ensemble of shots. See also answer to next comment.

Regarding the use of arbitrary units in the paper, we see the confusion. When a conversion to a probability scale was not possible or did not make sense, we have now added the correct units in the main paper and in Fig. S2c.

We note that during the experiment, the charge sensor characteristics changed many times and therefore the values are not fully consistent. However, the difference in current between the blocked and unblocked states remained in the range of 1 nA to 1.5 nA.

II.f. The previous point implies that the authors have NOT performed single-shot readout for any of the data. Why was this not done? Fig. S2c shows very good charge readout fidelity with two clearly separated histogram peaks and the last paragraph of the Supplementary Section B states that Fig. S2c shows the histograms of 1000-shots measurements and “the device could allow for operation in the single-shot readout regime”.

Thank you for the question.

We want to clarify that the stated current values were a result of the averaged time-resolved current with 1000 shots integrated directly on the digitizer. An approximate probability scale $\sim P$ for the different plots was calculated slightly differently for the different cases (as stated in the different captions).

Indeed the histogram in Fig. S2c reveals that single shot measurements would be possible in this device, but we indeed do not perform them. However, we are confident that the readout scheme used allowed us to fully reveal the physics discussed in the paper accurately. In the RB data, we intentionally use two different initial states and interleave the n_C axis. This method (used also by others before us) allows to avoid confusing charge sensor drift with a RB decay.

In a subsequent work from our lab, we extensively study the SPAM mechanisms and where the method for calculating P was indeed based on the histogram: E. G. Kelly et al., arXiv:2504.06898 (2025).

We add:

1)

for either an identity operation (I) or for an X_π gate on qubit 1. The resulting histogram, shown in Fig. S2c, reveals two well-separated Gaussian peaks corresponding to the $|\downarrow\downarrow\rangle$ and $|\uparrow\downarrow\rangle$ states, demonstrating that the device could allow for operation in the single-shot readout regime ~~—(as performed in Ref. [40])~~. The fidelity after applying a (1

2)

tinguishing $|\uparrow\downarrow\rangle$ from the other three states (see SI Sec. B). The ~~resulting~~ time-resolved sensor current is integrated over 100 μ s at the R point, and then averaged over 1000 experimental shots. The resulting average I_{sensor} is converted to an approximate probability scale \tilde{P} and the method used to calibrate the scale is explained in the captions of the figures of the respective measurements.

III. Resonantly-Driven ST Qubit:

III.a. There is a discrepancy between the direction of J_{AC} in the text and in the figure. In the text, J_{AC} is defined along σ_x , while in Fig. 1e it points along $-X$.

Thank you for pointing this out. We have corrected the figure to match the definition used in the text.

III.b. The fact that the applied, sinusoidal driving voltage gets translated into the exponential of a sinusoid, means that the system is not actually driven with a sinusoid as the authors point out in this section and in Fig. S3. Have the authors tried to compensate their driving waveform for this by applying a voltage proportional to $\log(\sin)$?

We appreciate the suggestion. We did not attempt to compensate the waveform using a $\log(\sin)$ shape. However, we note that such compensation is not strictly needed, as far as we can tell. See the similar discussion in response to Reviewer 1 question #6.

We now include a comment on this in the manuscript and reference related work by M. Rimbach-Russet al. Quantum Science and Technology 8, 045025 (2023)

simple to implement. Additionally, the drive signal on the barrier could be transformed to compensate for the exponential dependence of the exchange interaction on the voltage, suppressing the higher order terms [39].

III.c. The implementation of randomized benchmarking states that the “Clifford gate set $\{X_{\pi/2}, X_{\pi/2}, Z_{\pi/2}, Z_{\pi/2}, I\}$ ” is used. This definition is incorrect. The single qubit Clifford gate set consists of 24 different gates, and Clifford randomized benchmarking uses sequences that are randomly constructed out of these 24 gates. The paper by Epstein et al. is a good reference for this: <http://dx.doi.org/10.1103/PhysRevA.89.062321>. We recommend deleting the word “Clifford” here and as appropriate in the whole paragraph. The text also states that “ $Z_{\pi/2}, Z_{\pi/2}, I$ are considered instantaneous and error-free” as they are only implemented virtually. We understand the purpose of a virtual Z gate, but a virtual I gate is not doing anything at all and should just be omitted from the gate set. If an identity is included in the gate set, then it should be implemented as a proper idle gate with a certain duration (e.g., duration of $X_{\pi/2}$).

We apologize for the imprecise language. We meant to say that we generate the Clifford gate set using the named set of gates.

Regarding the ‘do-nothing-for-no-time’ I gate, we have included it in the gate set because we wanted to have full gate set and be consistent with RB sequences produced by Qiskit and include the I gate. In any case, this gate has not been counted as physical for both RBs in the updated version.

We further clarify the sequences used for the RB sequence of each qubit in newly added section in the supplementary information:

I. RANDOMIZED BENCHMARKING

A. Resonantly driven ST qubit

The Clifford gates for single-qubit RB are composed as per Tab. S2 [9, 40]. The average number of gates per Clifford is 2.125, and the percentage of physical gates is 0.392 (see Tab. S2). Average physical gate infidelity is calculated as $(1 - F_C)/2.125/0.392$, where F_C is the average Clifford gate fidelity. The provided 1σ uncertainty is computed with respect to the nonlinear least squares fit as the square root of the corresponding diagonal element of a covariance matrix (using `scipy.curvefit`).

B. Dressed ST qubit

The Clifford gates for single-qubit RB are composed as per Tab. S2 [9, 40]. The average number of gates per Clifford is 1.875, and the percentage of physical gates is approximately 0.98 (see Tab. S2). The average physical gate infidelity is calculated as $(1 - F_C)/1.875/0.98$.

Table S2. Clifford gate implementations (top) and physical gate fractions (bottom) for resonantly driven and dressed ST qubits. For the resonant qubit, there are 2.125 gates per Clifford on average and a 0.392 physical gate fraction; for the dressed qubit, 1.875 gates per Clifford and approximately 0.98 physical gate fraction.

Clifford Gates	
Resonant	Dressed
I	I
-Z, X	Y/2, X/2
-Z, X/2	-X/2, -Y/2
-Z, X/2, -Z	X
-Z, X/2, -Z/2	-Y/2, -X/2
-Z, X/2, Z/2	X/2, -Y/2
-Z/2	Y
-Z/2, X/2	-Y/2, X/2
-Z/2, X/2, -Z	X/2, -Y/2
-Z/2, X/2, -Z/2	X, Y
-Z/2, X/2, Z/2	Y/2, -X/2
X	-X/2, Y/2
X, Z/2	Y/2, X
X/2	-X/2
X/2, -Z	X/2, -Y/2, -X/2
X/2, -Z/2	-Y/2
X/2, Z/2	X/2
Z	X/2, Y/2, X/2
Z/2	-Y/2, X
Z/2, X	X/2, Y
Z/2, X/2	X/2, -Y/2, X/2
Z/2, X/2, -Z	Y/2
Z/2, X/2, -Z/2	-X/2, Y
Z/2, X/2, Z/2	X/2, Y/2, -X/2

Resonant Gate Composition		Dressed Gate Composition	
Gate Type	Percentage	Gate Type	Percentage
$\pm Z/2, \pm Z$	58.82%	$\pm Y/2$	35.6%
$\pm X/2$	31.37%	$\pm Y$	8.9%
$\pm X$	7.84%	$\pm X/2$	44.4%
I	1.96%	$\pm X$	8.9%
		I	2.2%

III.d. Related to the previous comment, in the caption of Fig. S1a, the authors state that they apply an identity gate for the measurement in the left panel. Was this actually an identity gate or did they just do nothing?

Thank you for the question. We confirm that the I gate, wherever is used in the paper, is a 'do-nothing-for-no-time' gate.

This is the case for RB of the dressed qubit where we had mistakenly counted it as a proper I (wait) gate. Upon revisiting the code we indeed found that was a 'do-nothing-for-no-time' gate as well. Please see reply of Referee 1 question 3.

However, in Fig S1a, the purpose of the identity gate is to calibrate the probability scale and characterize the readout. It has no impact on the RB results.

V. Dressed ST Qubit Spectroscopy:

V.a. Where do the additional features next to the center peak in Fig. 3a come from?

Are pump and probe coherent?

The authors state in the caption for Fig. S4 that "The phase of the second tone is shifted by $\pi/4$ to get better agreement with the experimental data...", which implies that there was a fixed relation between pump and probe beam.

It would be interesting for the readers to see how pump and probe beam are generated, which relates back to our previous comment about the missing description of the experimental setup.

We thank the reviewers for their question.

Regarding the setup and signal generation, we refer the reviewers to the newly added "**Experimental Set-up**" section. Both the pump and probe signals are generated from the same channel of the AWG, and the final waveform is simply the sum of the two.

In both the experiment and the simulations, the pump and probe signals have a phase difference of $\pi/4$. This choice was made to ensure that all three branches of the Mollow triplet (corresponding to spin rotations) are visible with comparable and satisfactory contrast. We also tested other phase differences (e.g., 0 and $\pi/2$), as well as various drive durations, but in all these cases, one of the branches consistently appeared significantly less pronounced than the others. It is beyond the scope of this paper to give intuition for this behavior.

Regarding the origin of the features near the center peak:

- If the reviewers are referring to the two additional main lines, these correspond to the transitions of the dressed spin, as discussed in the supplementary material.
- If the reviewers are referring to the smaller wave adjacent to the central transition, which is present in both experiment and simulation, we attribute this to non-resonant driving of the singlet-triplet qubit.

We will add in the caption of the figure that the two signals had a phase difference of $\pi/4$ to increase visibility of all 3 lines.

the measured Rabi oscillation pattern is disrupted. The two signals have a phase difference of $\pi/4$. This choice was made to achieve maximum visibility of all three branches. **b.** A frequency modulated driving signal gives rise to an oscillating (with

VI. Dressed ST Qubit Control:

VI.a. Fig. 3b-ii shows the dressed qubit Rabi frequency Ω_{ST}^{FM} as a function of the frequency modulation $\Delta\nu_{FM}/2$ up to 3 MHz. This is implemented for $\Omega_{ST} = 4.2$ MHz. For the fastest driving, the driving speed is 70% of the level splitting. We are a bit surprised that there are no effects of breakdown of the rotating wave approximation visible in the figure. Not even a fading of the oscillations towards higher driving speeds as described in Section E of the supplementary material. See Ref. 34 for comparison. Important here again is if there is a fixed phase relation between the RF sources.

We are grateful to the reviewers for this question which lead us to find an embarrassing mistake in the data analysis for this plot. We will first explain the raw acquired data and why we were confused. Then we will present the correct analysis supported by additional simulations.

A mistake in the data acquisition code mislabeled the x and y axis. In addition, the data are symmetric and taken until the value 10 when plotted in reasonable units μ s for time and MHz for drive bandwidth. Below we show the raw data as shown from the plotter inspector of qcodes.

Raw data as plotted from the QCodes tool Plotter Inspector

This led us to interpret the data transposed. Because of this, we made a mistake and **took the FFT in the wrong axis (in the df axis instead of the time axis)**. Below we plot the FFT along the correct and the wrong axis until $Df=3$ MHz as we did in the paper.

Indeed, as the referee suggests, there is now a slight fading away of the FFT amplitude.

In addition, we now plot below the FFT alongside the full values of Df , and we see that as the Df increases the fading of the FFT increases and additional frequencies appear revealing the collapse of the RWA. These features were also observed in Qutip simulations of the FM-driven system (left), which agree qualitatively with the measurements (right).

While the detailed study of the behavior of the system for large Df falls outside of the scope of this experiment we believe that they can be very interesting for future discussions and therefore we include them in a newly added section of the Supplementary information.

We change:

the linear 1:1 relationship between $\Delta\nu_{\text{FM}}/2$ and $\Omega_{\text{ST}}^{\text{FM}}$ [35]. By increasing $\Delta\nu_{\text{FM}}/2$ to values close to or larger than $\Omega_{\text{ST}}^{\text{FM}}$ (see SI Sec. F), the system enters a regime where the RWA ceases to be valid [35]. Additionally, in this scheme, increasing the drive amplitude does not necessitate a higher voltage, which could cause heating or crosstalk [36, 37], but rather ~~a broader bandwidth~~ by inserting a broader modulation bandwidth $\Delta\nu_{\text{FM}}$, limited only by the capabilities of the control electronics [18].

The Fig.3 bii of the manuscript (also made an adjustment to the inset) and the Fig.3b(ii) where $\Delta\nu=2$ MHz was label instead of $\Delta\nu=1$ MHz as correctly described in the text.

We added a section in SI:

F. DRIVE BANDWIDTH OF THE DRESSED ST QUBIT

We expand on the measurement shown in Fig. 3b(ii) of the main text, showing $\Omega_{\text{ST}}^{\text{FM}}$ as a function of $\Delta\nu_{\text{FM}}$. While in the main text we focused on drive bandwidths lower than $\Omega_{\text{ST}}^{\text{FM}}$ in Fig. S6 we show how the system behaves when driven up to $\Delta\nu_{\text{FM}}/2 = 5$ MHz. As can be seen from both the quantitative simulation performed by QuTiP and from the measured data for large values of $\Delta\nu_{\text{FM}}/2$, the RWA seems to start breaking as the FFT amplitude fades away and additional frequency components appear. Further study of the dynamics in this regime can be found in Ref. [35] but fall outside of the scope of this study, where we operate the dressed ST qubit at a maximum $\Delta\nu_{\text{FM}} \approx 1$ MHz.

Figure S6. Dressed ST qubit drive bandwidth. a. Dressed ST qubit driving for different values of $\Delta\nu_{\text{FM}}$: QuTiP simulations (left) and experimental measurements (right). b. The FFT along the time axis (QuTiP simulations (left) and experimental measurements (right)) revealing the dependence of the dressed Rabi frequency $\Omega_{\text{ST}}^{\text{FM}}$ on $\Delta\nu_{\text{FM}}$.

VI.b. Related to the previous comment, Supp. Section E states that “When driving with $\Delta\nu_{\text{FM}} = 1$ MHz, we observe that the oscillations begin to fade at higher modulation frequencies (f_{FM}), indicating the breakdown of the rotating wave approximation.”

Could the authors please add some extra details on this discussion? We can see that in Fig. S5 the oscillations for higher f_{FM} are more faded than for lower f_{FM} , but the reason for this is not clear. The authors describe this as “indicating the breakdown of the rotating wave approximation”, but a higher f_{FM} does not correspond to a higher driving speed, and in fact, a lower f_{FM} (at constant driving speed) brings the counterrotating term closer to the transition frequency.

We thank the reviewers again for their comment. We would like to clarify our statement. While the physical mechanism behind the fading of the oscillations at higher modulation frequencies is not fully intuitive to us, we observe that the effect becomes more pronounced for larger modulation amplitudes ($\Delta\nu_{\text{FM}} = 1$ MHz). This observation leads us to speculate that the phenomenon may be related to a breakdown of the rotating wave approximation (RWA), although we acknowledge that the connection is not fully established. Notably, QuTiP simulations (shown below) also reproduce this behavior.

We have added:

When driving with $\Delta\nu_{\text{FM}} = 1 \text{ MHz}$, we observe that the oscillations begin to fade at higher modulation frequencies (f_{FM}), ~~indicating the a phenomenon that may be related to a~~ breakdown of the ~~rotating-wave approximation~~RWA. However, this effect does not impact the resonant drive at $f_{\text{FM}} = 4.2 \text{ MHz}$. By introducing an additional phase

VI.c. The sentence “Additionally, in this scheme, increasing the drive amplitude does not necessitate a higher voltage, which could cause heating or crosstalk [35, 36], but rather a broader bandwidth, limited only by the capabilities of the control electronics [18].” can be made clearer, e.g. by inserting “a broader modulation bandwidth $\Delta\nu_{\text{FM}}$,...”.

Yes, we adjust the text:

[36, 37], but rather ~~a broader bandwidth~~by inserting a broader modulation bandwidth $\Delta\nu_{\text{FM}}$, limited only by the capabilities of the control electronics [18].

VI. d. The randomized benchmarking of the dressed ST qubit is also done with a set of 5 qubit gates. This time the text states that “that all gates in the set are physical”. Does this include the identity gate?

After revisiting the measurement code we found that we had mistakenly put the I gate as a physical (wait $\pi/2$) gate. In reality, all I gates used anywhere in the experiment are ‘do nothing for no time’ gates. We thank the reviewers for spotting the discrepancy and we adjust. The impact on the gate fidelity is within the error margin.

ford, and noting that all gates except of the I^{FM} (do nothing for no time) in the set are physical. This fidelity is, within error bars, comparable to that of the resonant ST qubit gates.

Supplementary Section A:

S.a. The way this section reads makes it hard to understand which state is blockaded and which is not. In Fig. S1a (and the first paragraph of the section where this figure is explained), the down-down state seems to be unblockaded. On the other hand, in Fig S1d (and the third paragraph) states that the down-down state is blockaded. Then only Fig. S1e shows that the blockaded state is changing depending on the $t_{\text{ramp_out}}$, indicating that the $t_{\text{ramp_out}}$ in Fig. S1a must also be $1\mu\text{s}$. To minimize confusion, we suggest adding a pulse diagram to Fig. S1a and replace the sentence “(discussed in detail in the next paragraph)” with “(discussed in detail in the third paragraph)”.

We thank the reviewer for their comments. We understand the confusion, as we first presented the final PSB readout result in Fig. S1a, followed by a detailed explanation of the tuning procedure that led to this outcome. However, we believe this sequence is helpful: readers primarily interested in the latched windows can immediately see the final result, while those seeking a deeper understanding of the tuning steps can follow the subsequent explanation.

Regarding the question on the blockaded state between **Fig S1a** and **Fig S1d**. The reviewer has correctly understood that the two plots show different states to be unblocked. This is however exactly what we expect, as in Fig S1d the return ramp is zero and therefore the anticrossing is crossed diabatically. Therefore the previous unblocked state $\downarrow\downarrow$ is now blocked. We state this in paragraph 3 ‘Finally, we note that the rapid return pulse to the interdot results in the diabatic crossing of the $S(2, 0)$ - $T(1, 1)$ anticrossing for all states. This maps the $\downarrow\downarrow$ state to the $(2, 1)$ charge configuration instead.’

For Fig S1a we add in the caption:

The ramp time between points $1 \leftrightarrow 2$ was set at $t_{\text{ramp}} = 1 \mu\text{s}$

S.b. In the second last paragraph of the Section A, only Fig. S1d is referred to while discussing the chirped pulse experiment. However, that panel does not involve any chirped pulse. Should the discussion refer to the Fig. S1e instead? Similarly, the last paragraph should refer to the Fig. S1f.

We apologize, this was a typo. We have now corrected the references to match the figure numbers.

Reviewer #3 (Remarks to the Author):

The authors report the resonant driving of a singlet-triplet qubit in germanium by modulating the exchange coupling. Furthermore, by continuously driving the qubit they demonstrate the dressed singlet-triplet qubit with an increased $T2^*$ of 20.3 μ s. The reported single qubit gate fidelity of 99.68% is among the highest reported values in literature. Overall, the paper is well-written, the storyline is clear, and the measurement data is of high quality. I will be happy to support its publication in Nature Communications after the authors help me clarify a few points in the manuscript.

1) The dressed qubit shows a ten-fold increase in $T2^*$ compared to the bare qubit. However, the gate fidelity is not higher. In that case, does the dressed qubit have advantages over the bare qubit?

See answer in Reviewer 1 question 9.

2) What are the limiting factors for gate fidelities of the bare qubit and the dressed qubit, respectively?

See answer to the second part of Reviewer 1 question 3.

3) In fig 3(c), the authors use normalized I_{sensor} to fit $T1$ and $T2_{\text{Hahn}}$. The measurement noise seems quite high compared to other data. Why do they plot fig 3(c) using I_{sensor} instead of the state population as in other figures?

As also indicated from the Question II.e of Reviewer 2, the data in this figure part are now plotted in the original measured I_{sensor} . We did not normalize to P here because no prepend pulses were taken for the two experiments to set the normalization limits.

Regarding the noise level, it is indeed higher in these experiments; we attribute it to lower visibility due to the fact that both initialization and readout include an extra resonant ST $Y/2$ pulse (which we did not optimize as we did for the RB experiment) together with the fact that we are continuous driving for long durations which can induce further instability of the charge sensor.

4) When characterizing the dressed qubit fidelity the authors use a gate set which is different from the set they use for the bare singlet-triplet qubit. What is the reason behind this choice? I suppose using the same gate set would make better one-to-one comparison?

See answer to Reviewer 1 question 3.

Reviewer #4 (Remarks to the Author):

Acknowledged, no response required.